# Curcumin improves D-galactose and normal-aging associated memory impairment in mice: In vivo and in silico-based studies

Md. Ashrafur Rahman[1,2]*, Arif Anzum Shuvo[1], Asim Kumar Bepari[1], Mehedi Hasan Apu[1], Manik Chandra Shill[1], Murad Hossain[1], Mohammed Uddin[3,4], Md. Rabiul Islam[5], Monjurul Kader Bakshi[1], Javed Hasan[1], Atiqur Rahman[1], Ghazi Mohammad Sayedur Rahman[1], Hasan Mahmud Reza[1]*

1 Department of Pharmaceutical Sciences, North South University, Bashundhara, Dhaka, Bangladesh, 2 Department of Pharmaceutical Sciences, Jerry H. Hodge School of Pharmacy, Texas Tech University Health Science Center (TTUHSC), Amarillo, TX, United States of America, 3 College of Medicine, Mohammed Bin Rashid University of Medicine and Health Sciences, Dubai, UAE, 4 Cellular Intelligence (Ci) Lab, GenomeArc Inc., Toronto, ON, Canada, 5 Department of Pharmacy, University of Asia Pacific, Dhaka, Bangladesh

* ashrafurrahman82@gmail.com (MAR); hasan.reza@northsouth.edu (HMR)

**Data Availability Statement:** We have uploaded a file (defined as Data set_ Curcumin_Plos one) in

## Abstract

Aging-induced memory impairment is closely associated with oxidative stress. D-Galactose (D-gal) evokes severe oxidative stress and mimics normal aging in animals. Curcumin, a natural flavonoid, has potent antioxidant and anti-aging properties. There are several proteins like glutathione S-transferase A1 (GSTA1), glutathione S-transferase omega-1 (GSTO1), kelch-like ECH-associated protein 1 (KEAP1), beta-secretase 1 (BACE1), and amine oxidase [flavin-containing] A (MAOA) are commonly involved in oxidative stress and aging. This study aimed to investigate the interaction of curcumin to these proteins and their subsequent effect on aging-associated memory impairment in two robust animal models: D-Gal and normal aged (NA) mice. The aging mice model was developed by administering D-gal intraperitoneally (i.p). Mice (n = 64) were divided into the eight groups (8 mice in each group): Vehicle, Curcumin-Control, D-gal (100mg/kg; i.p), Curcumin + D-gal, Astaxanthin (Ast) + D-gal, Normal Aged (NA), Curcumin (30mg/kg Orally) + NA, Ast (20mg/kg Orally) + NA. Retention and freezing memories were assessed by passive avoidance (PA) and contextual fear conditioning (CFC). Molecular docking was performed to predict curcumin binding with potential molecular targets. Curcumin significantly increased retention time (p < 0.05) and freezing response (p < 0.05) in PA and CFC, respectively. Curcumin profoundly ameliorated the levels of glutathione, super-oxide dismutase, catalase, advanced oxidation protein products, nitric oxide, and lipid peroxidation in mice hippocampi. In silico studies revealed favorable binding energies of curcumin with GSTA1, GSTO1, KEAP1, BACE1, and MAOA. Curcumin improves retention and freezing memory in D-gal and nature-induced aging mice. Curcumin ameliorates the levels of oxidative stress biomarkers in mice. Anti-aging effects of curcumin could be attributed to, at least partially, the upregulation of antioxidant enzymes through binding with GSTA1, GSTO1, KEAP1, and inhibition of oxidative damage through binding with BACE1 and MAOA.

the supporting information section containing all data sets of our manuscript.

**Funding:** This work was partially supported by the research grant provided by the CTRG-North South University 2019–2020 and the National Institute of Science and Technology (NST- 2020). The funders had no role in study design, data collection, analysis, decision to publish, or manuscript preparation. No additional external funding was received for this study.

**Competing interests:** Author Mohammed Uddin is employed by Cellular Intelligence (Ci) Lab, GenomeArc Inc., Toronto, ON, Canada. The remaining authors declare that the research was conducted in the absence of any commercial or financial relationships that could be construed as a potential conflict of interest.

## 1. Introduction

Aging is a natural process characterized by gradual deterioration in diverse physiological functions [1], including oxidative damage-driven memory loss [2]. Memory dysfunction could be triggered by an imbalance among reactive oxygen species (ROS), reactive nitrogen species (RNS), and antioxidant enzyme activities [3]. The excess generation of ROS, RNS, and reduction in antioxidant enzyme activities in the brain escalate lipid peroxidation, protein oxidation, and mitochondrial DNA (mtDNA) damage [4], contributing to brain aging [5]. Brain aging and accompanied cognitive dysfunction are typical features of neurodegenerative diseases [6]. One study claimed that D-galactose (D-gal) accelerates the brain-aging process in animal models similar to the normal aging in humans [7]. D-gal-induced aging mice model has gained popularity among researchers because of its feasibility, fewer side effects, and higher survival rate of animals. It was reported that D-gal evokes aging-induced memory impairment through neurodegeneration, aberrant immune responses, and abnormal gene expression [8]. Another study suggests that D-gal increases malondialdehyde (MDA) level and decreases the antioxidant enzymes such as superoxide dismutase (SOD) and glutathione (GSH) [9]. At high doses, D-gal can accelerate cellular ROS generation through multiple mechanisms, such as adenosine triphosphate (ATP) depletion, redox homeostasis impairment, and elevation of the advanced glycation end product (AGE), the receptor for the advanced glycation end product (RAGE), and nicotinamide adenine dinucleotide phosphate (NADPH) oxidase [10]. Overproduction of ROS induces oxidative stress, cellular apoptosis, inflammation, and mitochondrial dysfunction, which leads to neuronal degeneration [11]. Raised free radical levels were implicated in cholinergic neuron dysfunctions in the brain [12] and aging-associated memory impairment [13]. Currently, several acetyl-cholinesterase (AChE) inhibitors (donepezil, rivastigmine, and galantamine) and an N-Methyl-D-aspartate (NMDA) receptor antagonists (memantine) are used in aging-associated memory impairment. The outcome of the treatment has not reached the optimum level owing to side effects and cost-ineffectiveness. Studies showed that natural compounds could be a promising solution for aging-associated memory impairment due to their antioxidants, anti-inflammatory, and anti-aging properties [14]. Furthermore, plant-derived compounds can be a choice on account of their cost-effectiveness. Curcumin, a natural compound found in Curcuma Longa, is regularly used as spicey ingredients and possesses anti-inflammatory, antioxidant, and anti-aging properties [15] by regulating several proteins such as tumor necrosis factor alpha (TNFα) [16], mammalian target of rapamycin (mTOR), sirtuin, and adenosine monophosphate-activated protein kinase (AMPK) [17]. However, several proteins like glutathione S-transferase A1 (GSTA1), glutathione S-transferase omega-1 (GSTO1), kelch-like ECH-associated protein 1 (KEAP1), beta-secretase 1 (BACE1), and amine oxidase [flavin-containing] A (MAOA) are commonly involved in oxidative stress and aging [18]. GSTA1 regulates GSH conjugation, which may reduce oxidative stress [19]. Another protein, GSTO1, modulates conjugation of GSH [19], the activation of interleukin-1β, and inflammation in aging-associated neurodegenerative disease [20]. KEAP1 plays an antioxidative role by regulating the Nrf2 cytoprotective signaling pathway [21]. BACE1 is closely associated with the generation of β-amyloid (Aβ) in aging-induced neurological disease, Alzheimer's Disease [22]. Overproduction of MAOA in the brain triggers oxidative overload [23]. However, no study has detected the interaction of curcumin with antioxidant and aging regulating proteins GSTA1, GSTO1, KEAP1, BACE1, MAOA and their consequent impact on improving the memory. Therefore, it would be interesting to find the binding affinity of curcumin with these proteins in order to determine their subsequent impacts on modulating the oxidative biomarkers in the aging process. We have extensively

investigated the effects of curcumin on two robust animal models; the normal aging and the D-gal induced aged mice model, using two widely acceptable behavioral tasks: passive avoidance (PA) and contextual fear conditioning (CFC). We also biochemically assayed the levels of oxidative stress biomarkers, including glutathione (GSH), superoxide dismutase (SOD), catalase (CAT), advanced oxidation of protein products (AOPP), nitric oxide (NO), and malondialdehyde (MDA) in mice hippocampi. Moreover, we performed in silico analysis to illustrate the binding of curcumin with protein molecules implicated in brain oxidative stress pathophysiologies. Overall, this study has comprehensively discerned the effects of curcumin on aging-related memory impairment by a battery of behavioral, biochemical, and molecular docking experiments.

## 2. Materials and methods

### 2.1. In vivo

**2.1.1. Chemicals.** Curcumin, D-gal, Ast, and thiobarbituric acid (TBA) trichloroacetic acid (TCA) were purchased from Sigma-Aldrich (Germany). All other chemicals, reagents, and solvents used in this study were analytical grade.

**2.1.2. Experimental animals.** We used randomly selected healthy male *Swiss albino* mice in this study. We chose male mice because of their tendency in increasing context fear expression toward adulthood, unlike female mice [24]. Forty adult male mice (40 ± 2 gm; 6–8 weeks) were divided into four groups, and 24 normal-aged mice (42 ± 2 gm; 10–12 months) were divided into three groups. Each group consisted of eight mice. Mice were reared at 25˚C temperature, 55% ± 15% relative humidity, 12:12 hours light-dark cycle with an adequate supply of food pellets and pure drinking water.

**2.1.3. Preparation of drugs.** Curcumin was suspended in 0.25% w/v sodium carboxymethylcellulose and administered at a 1ml/100g body weight [25]. D-gal was prepared freshly by dissolving in 0.9% of saline before each session [26]. Ast powder was dissolved in distilled water at the ratio of 20mg/20ml [27].

**2.1.4. Experimental design.** We divided sixty-four mice into the following eight groups (Each group consisted of eight mice):

1. Vehicle (n = 8): 0.25% w/v sodium carboxymethylcellulose was administered (oral gavage) [25] once daily for ten weeks.

2. Curcumin-Control (Positive Control; n = 8): Curcumin 30mg/kg [25] was administered orally (oral gavage) once daily for ten weeks.

3. D-gal (n = 8): D-gal (100mg/kg) was administered intraperitoneally [26] once daily for ten weeks.

4. Curcumin + D-gal (n = 8): Curcumin 30mg/kg [25] and D-gal 100mg/kg [26] were administered orally (oral gavage) and intraperitoneally, respectively, once daily for ten weeks.

5. Ast (standard antioxidant) + D-gal (n = 8): Ast 20mg/kg [28, 29] and D-gal 100mg/kg [26] were administered orally (oral gavage) and intraperitoneally, respectively once daily for ten weeks.

6. NA (n = 8): Normal mice pellets and water was given daily throughout the experiment.

7. Curcumin + NA (n = 8): Curcumin 30mg/kg was given orally (oral gavage) [25] once daily for ten weeks.

8. Ast (standard antioxidant) + NA (n = 8): Ast (20mg/kg) was administered orally (oral gavage) [28, 29] once daily for ten weeks.

After ten weeks of treatment, we investigated retention and freezing memory by PA and CFC tests, respectively. The animals were then euthanized for collecting brain tissues to assay oxidative stress biomarkers. Finally, the molecular targets of curcumin were assessed by in silico analysis (Fig 1).

**2.1.5. Passive avoidance test.** The method of PA was performed as demonstrated by Tabrizian [30]. The PA behavior study was executed with a step-through type avoidance learning task that calculates the memory retention of mice. This test consisted of three sessions. Firstly, each mouse was placed gently in the experimental apparatus in the habituation session and allowed to habituate for at least 5 minutes on day 1. Mice that did not enter into the dark chamber after keeping them in a light chamber for more than 120 s were excluded from the experiment. Secondly, the training session was used to evaluate the learning and exploration. This session consisted of 3 trials with 30 min intervals. In the first trial, each mouse was gently placed in the light, and the sliding door was opened after 10 seconds. Each mouse was allowed to explore both chambers for 5 minutes. Each mouse's time before entering the dark chamber with all four paws was recorded as the retention period (300 s was considered a cut-off point). The second trial was performed in a similar way to the first trial. In the third trial, once a mouse stepped through the dark chamber with all four paws, the sliding door was closed. An electrical foot shock (0.3–0.5 mA for 2 s) was delivered. After the foot shock, each mouse was kept in the dark chamber for an additional 10 s to build an association between chambers and the foot shock, and then the mouse was returned to the home cage. Repeated training was conducted similarly for five consecutive days with 24 hours intervals. The objective of this testing session was to determine memory RT after 24 and 48 hours of training. No electrical foot-

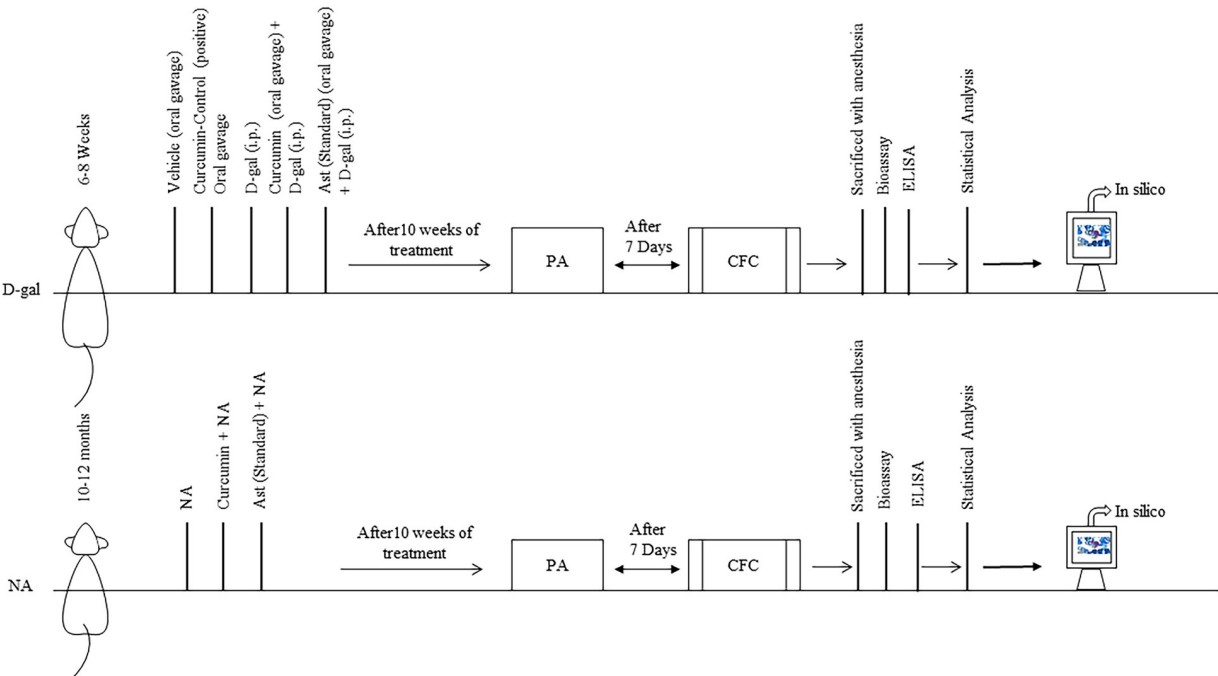

**Fig 1. Schematic diagram of total experimental procedures: The treatment was continued for ten weeks.** After that, the behavioral tests were performed using Passive Avoidance (PA) and Contextual Fear Conditioning (CFC). The biomarkers were detected after completing the behavioral tasks. In-silico studies were used to determine curcumin's binding affinity to targeted proteins.

shock was applied during this session. The retention period was measured using a stopwatch. The apparatus was adequately cleaned with 70% ethanol before the experiment.

**2.1.6. Contextual fear conditioning test.** The CFC Test was performed after seven days of completion of the PA test. The method of CFC was performed as demonstrated by Shoji [31]. In brief, each mouse was placed in a transparent acrylic chamber ($33 \times 25 \times 28$ cm) with a stainless-steel grid floor (0.2 cm diameter, spaced 0.5 cm apart; O'Hara & Co., Tokyo, Japan) and was allowed to navigate independently for 2 min. Next, the conditioned stimulus (CS) of 55 dB white noise was presented for 30 s. During the last 2 s of CS presentation, the unconditioned stimulus (US) of a mild foot shock (0.3 mA, 2 s) was applied. Two more CS-US pairings were presented with a 2-min inter-stimulus interval. A context test was conducted to detect contextually conditioned fear memory in the same chamber approximately 24 hours (2a) and 30 days (31a) after the conditioning session. A cued test was performed to detect a novel fear memory in an altered context. The cued test was conducted in a triangular chamber ($33 \times 29 \times 32$ cm) made of opaque white plastic located in a different room. The test was done a few hours after the context test on day 2 (2b) and day 30 (31b). A computer-based infrared video system (Med Associates, Inc. USA) was used to monitor the freezing in mice [32].

**2.1.7. Tissue processing.** Eight representative mice data of each group were used to perform the biochemical analyses. Mice were anesthetized by administering 200μl of ketamine (50 mg/ml, Incepta Pharmaceuticals Ltd., Bangladesh) intraperitoneally. Mice were sacrificed by decapitation. The brain was extracted from the skull and transferred immediately to Petri dishes placed over ice. The hippocampal tissue was microdissected and immediately stored at –80°C. On the next day, the homogenate of hippocampal tissue 10% (w/v) was prepared in sodium phosphate buffer (1× PBS pH 7.0) supplemented with 1:100 protease inhibitor cocktail (Sigma, Saint Louis, MO, USA) by using Ultra-Turrax T25 (United States) homogenizer. Sonication of homogenized tissue was performed at a 5-s cycle for 150 s using an ultrasonic processor. The homogenized tissue was centrifuged at 10,000 rpm (RCF 11200) for 10 min at 4°C. The clear supernatants were diluted with 0.1x PBS buffer and performed the biochemical analysis.

**2.1.8. Oxidative stress measurement.** All determinations were normalized by the protein concentration of the samples. Total protein content was measured by Lowry's method [33].

*2.1.8.1. Estimation of Glutathione level.* GSH level was detected according to the previous method [34, 35]. Briefly, 2.7 ml of phosphate buffer (0.1 M, pH 8) and 0.2 ml of 5, 5-dithiol-bis (2-nitrobenzoic acid) were added with 1 ml of hippocampal homogenate. The color progressed was determined instantaneously at 412 nm. Results were expressed in μmol/mg protein.

*2.1.8.2. Determination of superoxide dismutase activity.* The SOD level was estimated based on a modified previous protocol [36, 37]. In short, the reaction mixture carried 50 mM sodium phosphate (pH 7.8), 13 mM methionine, 75 mM nitroblue tetrazolium (NBT), 2 mM riboflavin, 100 mM EDTA and 2 mL of hippocampal tissue homogenate. The change in absorbance of each sample was then documented at 560 nm after forming the blue formazan. The activity of SOD was expressed in U/30sec.

*2.1.8.3. Measurement of catalase activity.* The CAT activity was measured based on a previous method spectrometrically at a wavelength of 240 nm [38]. The reaction mixture (1.5 ml) comprised 1.0 ml of 0.01 M phosphate buffer (pH 7.0), 0.1 ml of hippocampal tissue homogenate, and 0.4 ml of 2 M $H_2O_2$. The reaction was stopped by adding 2.0 ml of dichromate-acetic acid reagent (5% potassium dichromate and glacial acetic acid were mixed in a 1:3 ratio). The activity of catalase was expressed in μmol/min/mg protein.

*2.1.8.4. Determination of advanced oxidation protein products.* AOPP was detected spectrophotometrically using a previous protocol [39, 40]. Briefly, 50 μl of hippocampal tissue homogenate was diluted with phosphate-buffered saline (PBS) at a ratio of 1:2. Chloramine T (0–100

mmol/L) was used for preparing the calibration curve. PBS was used as a blank. 100 μl of 1.16 M potassium iodide and 50 μl of acetic acid were added to each well, and absorbance at 340 nm was determined subsequently. Concentrations of AOPP were represented in chloramine units (μmol/ml).

*2.1.8.5. Estimation of nitric oxide level.* NO level was estimated according to a previous protocol [41] using the Griess-Illosvoy reagent. Griess–Illosvoy reagent was modified by using naphthyl ethylene diamine dihydrochloride (NED) (0.1% w/v) instead of 1-napthylamine (5%). The hippocampal tissue homogenates, phosphate buffer saline (0.5 mL), NED (1 mL), and sulfanilamide (1 mL) were diluted with PBS (2:8 ratio) and incubated at 25˚C for 15 min in a 96-well plate [35]. The absorbances were measured at a wavelength of 540 nm against the blank readings of the spectrophotometer. The concentration of NO was expressed in mmol/mg.

*2.1.8.6. Measurement of malondialdehyde level.* MDA was detected through colorimetric analysis by determining thiobarbituric acid reactive substances (TBARS) according to a previous protocol [42]. Briefly, 0.1 ml of hippocampal tissue homogenate in Tris–HCl buffer (pH 7.5) was treated with 2 ml of TBA-TCA-HCl (1:1:1 ratio) reagent (thiobarbituric acid 0.37%, 0.25 N HCl, and 15% TCA) and put in a water bath at 70˚C for 15 min and cooled. The absorbance of the clear supernatant was estimated against the reference blank at 535 nm [43]. The MDA level was detected by using a standard curve and represented in nmol/ml.

## 2.2. In silico (molecular docking)

**2.2.1. Prediction of molecular target.**   We predicted the molecular targets of curcumin from the literature search. We also used the SwissTargetPrediction (http://www.swisstarget prediction.ch) webserver [44].

**2.2.2. Preparation of protein structures.**   We downloaded 3D crystallographic structures of target proteins from the Protein Data Bank (https://www.rcsb.org/). Information on the pdb structures is given in Table 1. Protein structures were cleaned in UCSF Chimera [45]. We removed all non-amino acid residues and kept a single chain. The monomeric chain was then subjected to the Dock Prep module of UCSf Chimera, where any incomplete side chains were replaced using the Dunbrack rotamer library. The output from this step was saved as a pdb file. We next converted the pdb file to a pdbqt file using AutoDockTools [46]. Polar hydrogen atoms were added during this step, nonpolar hydrogens were merged, and the Kollman charges were added.

**2.2.3. Preparation of ligand structures.**   Curcumin's (PubChem CID: 969516) canonical slimes and 2D structure (as sdf) were obtained from PubChem (https://pubchem.ncbi.nlm.nih.gov/). We checked the accompanying literature for protein crystal structures, and Co-crystallized reference ligands were saved as pdb files in UCSF Chimera. All ligand structures were processed and saved as pdbqt files using Open babel [47] and POAP [48]. In short, ligands were optimized using the MMFF94 force field, all hydrogen atoms were added, 3D structures were generated, and energy minimization was done using 5000 steps of the conjugate algorithm.

**Table 1. Information of protein structures.**

| Protein Name | Common Name | UniProt ID | PDB ID | Reference Ligand (Co-crystallized) | Reference |
|---|---|---|---|---|---|
| Beta-secretase 1 | BACE1 | P56817 | 2QP8 | SCH734723 | [86] |
| Glutathione S-transferase A1 | GSTA1 | P08263 | 4HJ2 | Chlorambucil | [87] |
| Glutathione S-transferase omega-1 | GSTO1 | P78417 | 4YQM | C1-27 | [88] |
| Kelch-like ECH-associated protein 1 | KEAP1 | Q14145 | 6TYM | 08A | [89] |
| Amine oxidase [flavin-containing] A | MAOA | P21397 | 2Z5X | Harmine | [90] |

**2.2.4. AutoDock Vina molecular docking.** We performed Vina molecular docking [46] using the virtual screening tool of POAP. The grid box size was 24x24x24 angstroms, and the box was centered on the co-crystallized ligand in the pdb structure. The exhaustiveness was 8 for Vina docking. We recorded the estimated binding energy values for the best-docked poses of the ligands.

*2.2.4.1. Protein-ligand interactions.* From Vina docking simulations, we retrieved pdb files of the complexes with the best-docked poses (lowest estimated binding energy). Protein-ligand interactions were visualized using Discovery Studio Visualizer (BIOVIA, Dassault Systèmes, Discovery Studio Visualizer, v20.1.0.192, 2019). We compared the interactions of the co-crystallized ligand in the pdb file with that of the curcumin.

*2.2.4.2. 3D-rendering of protein-ligand complexes.* We used open-source Pymol [49] to render 3D images of ligands and the binding sites in the proteins.

## 2.3. Statistical analysis

A One-Way ANOVA followed by the Brown-Forsythe test was performed to determine the effect of the treatment groups. All analyses were executed by the GraphPad prism software. The difference was considered significant when the p-value was less than 0.05. Data were represented as mean ± SEM (Standard Error of the Mean).

## 2.4. Ethics

The institutional animal care and use committee (IACUC) of North South University (2019/OR-NSU/IACUC- No.0903) approved the study protocol. All the experimental procedures were performed following the NIH Guide for the care and use of laboratory animals. Throughout the experiments, all efforts were given to minimize the number of animals and to optimize their comfort.

# 3. Results

## 3.1. Effects of curcumin on retention time by adopting passive avoidance task

After 24 hours of training, the Vehicle and Curcumin-Control (Cur-Con) groups displayed mean retention times (RTs) of 210.93 ± 6.33 s and 276.31 ± 8.51 s, respectively (Fig 2). The RT was 116.43 ± 2.62 s in the D-gal injected mice group, indicating a significant decrease compared with the Vehicle ($F_{1, 14}$ = 2.07, p < 0.0001) and the Cur-Con ($F_{1, 14}$ = 2.17, p < 0.0001) groups (Fig 2). Contrarily, the decreasing trend of RT was prominently protected in the Curcumin + D-gal mice group (255.62 ± 14.11 s) (Fig 2). A One-Way ANOVA followed by the Brown-Forsythe test demonstrated a statistically significant difference in RT values between the Curcumin + D-gal and the D-gal injected mice groups ($F_{1, 14}$ = 88.31, p < 0.0001). An RT value of 249.87 ± 8.97 s was observed in the Ast + D-gal mice, which was significantly higher than the D-gal treated mice group ($F_{1, 14}$ = 7.95, p > 0.05; Fig 2). A similar tendency was observed in NA mice. The RT was 132.12 ± 3.11 s in the NA mice group (Fig 2). On the contrary, compared to the NA mice group, there was a significant protection from the decreasing trend of RT observed in the Curcumin + NA (247.25 ± 10 s; $F_{1, 14}$ = 31.10, p < 0.0001) and Ast + NA (269.81 ± 8.02 s; $F_{1, 14}$ = 2.06, p < 0.0001) mice groups (Fig 2).

Similarly, after 48 hours of training, the Vehicle and Cur-Con groups exhibited RTs of 202.43 ± 3.99 s, 271.87 ± 5.41 s, respectively (S1 Fig). As expected, RT values were considerably low in the D-gal injected mice (105.87 ± 3.58 s) and significantly high both in the Curcumin + D-gal (246.81 ± 14.38 s) and the Ast + D-gal mice (239.75 ± 10.04 s) (S1 Fig). Comparables

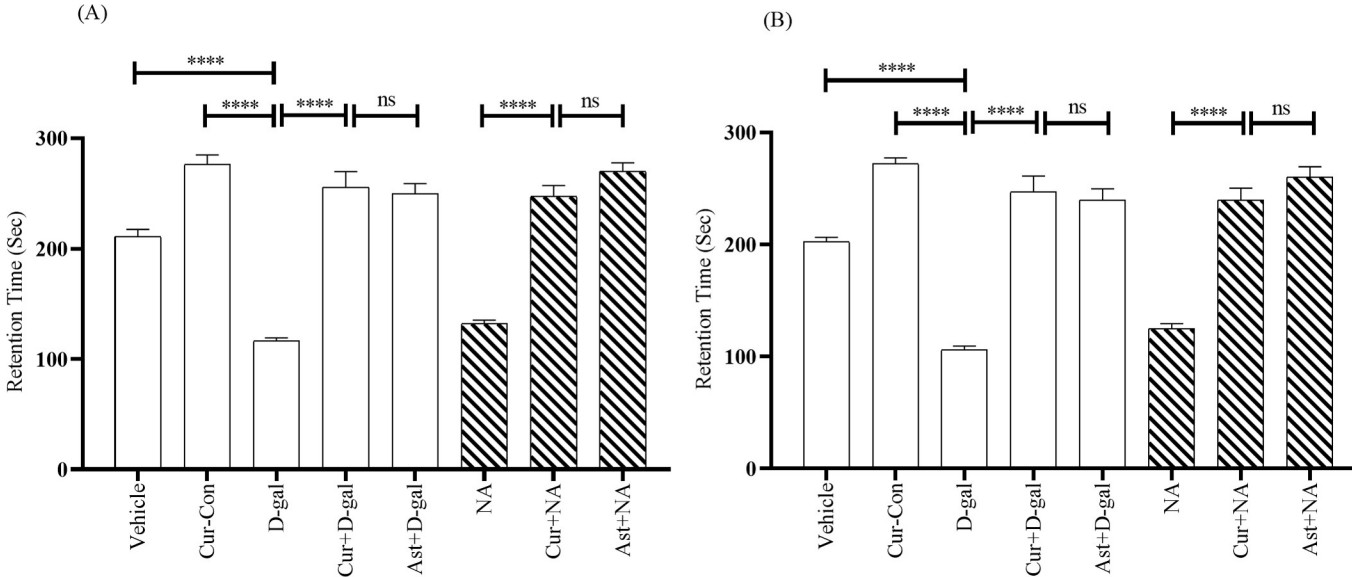

**Fig 2. Effect of curcumin on RT in D-gal and NA mice group after 24 hours of training.** The RT was calculated by performing PA tasks among Vehicle, Cur-Con, D-gal, Curcumin + D-gal, Ast + D-gal, NA, Curcumin + NA, Ast + NA groups. RT was expressed in second. Data was presented as mean ± SEM, n = 8 each group; ****p < 0.0001, ns = not significant.

RTs were also observed in the NA (125.06 ± 4.33 s), the Curcumin + NA (239.81 ± 10.38 s), and the Ast + NA (260.31 ± 9.16 s) groups.

### 3.2. Effects of curcumin on freezing response by performing contextual fear conditioning task

**3.2.1. Effects of curcumin on freezing response in conditioning session.** The contextual fear conditioning test was performed to evaluate the fear memory. On day 2(2a) of conditioning, the baseline activity was evaluated during the first 2 minutes in the novel environment. In this session, freezing response (FR) was detected without presenting the conditioned stimulus (CS, white noise) or unconditioned stimulus (US, foot-shock). The baseline activity was almost similar among Vehicle, Cur-Con, D-gal, Curcumin + D-gal, and Ast + D-gal groups (Fig 3A). During the last 6 min with CS-US pairings, the Vehicle and Cur-Con groups exhibited an FR of 61.87 ± 3.45%, 71.45 ± 3.17%, respectively (Fig 3A). The D-gal administered mice displayed an FR, 42.5 ± 3.60%, indicating a significant decrease compared with the Vehicle ($F_{1, 14} = 0.39$, p < 0.001) and the Cur-Con ($F_{1, 14} = 0.52$, p < 0.0001) groups (Fig 3A). Intriguingly, the decreasing tendency of FR was significantly protected in the Curcumin + D-gal mice group (67.60 ± 2.62%; $F_{1, 14} = 0.76$, p < 0.0001; Fig 3A) and similar trend was detected in the Ast (standard antioxidant drug) +D-gal mice group (66.56 ± 1.77%; $F_{1, 14} = 1.96$, p > 0.05). Likewise, the baseline activity was almost similar among the NA, Curcumin + NA, and Ast + NA groups (Fig 3B). During the last 6 minutes with CS-US pairing, the NA group exhibited an FR of 45.62 ± 0.82% (Fig 3B). Contrarily, compared to the NA mice group, the declining trend of FR was remarkably protected in the Curcumin + NA (64.79 ± 2%; $F_{1, 14} = 10.41$, p < 0.0001) mice group (Fig 3B). A similar change in FR was observed in the Ast (standard antioxidant drug) +NA mice group (63.95 ± 2.88%; p > 0.05; Fig 3B).

On day 31(31a) of conditioning, the Vehicle and Cur-Con groups displayed an FR of 55.83 ± 3.53%, and 66.04 ± 2.51%, respectively, during the last 6 min with CS-US pairings (Fig 3A). The FR was 35 ± 2.45% in the D-gal administered mice group, illustrating a significant

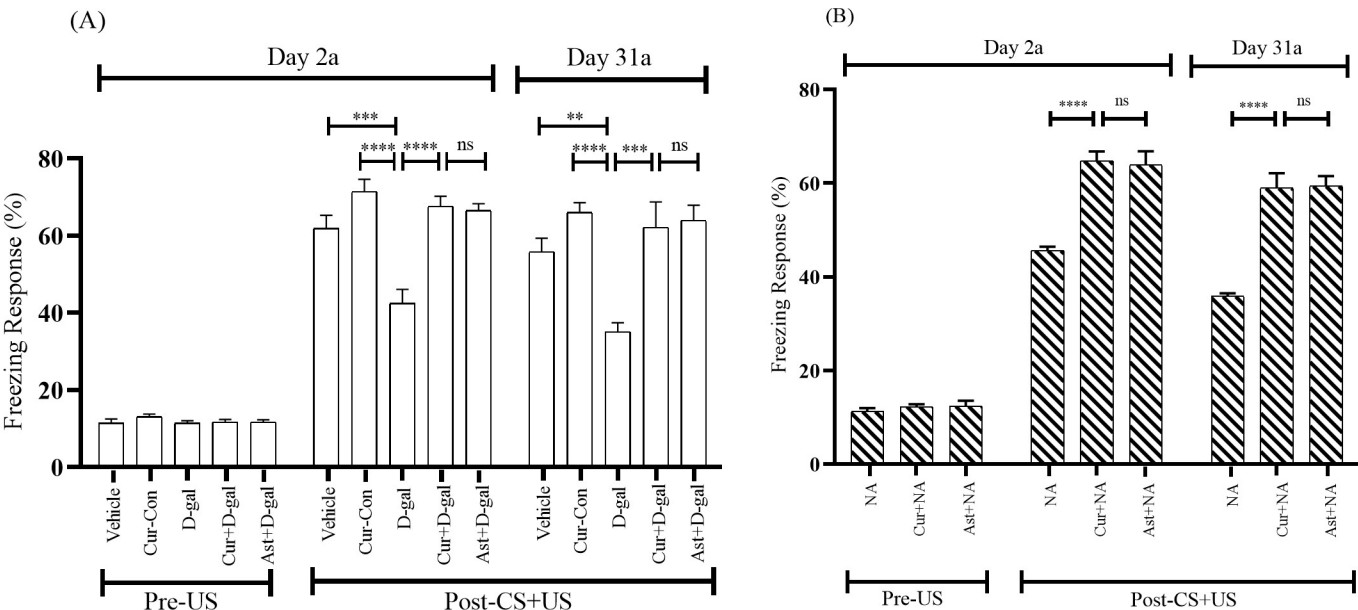

**Fig 3.** Effect of curcumin on the conditioning fear memory of (A) D-gal and (B) NA mice group. The memory was assessed by analyzing the FR. The FR was expressed in percentage (%). Data was presented as mean ± SEM, n = 8 each group; $^{**}p < 0.01$, $^{***}p < 0.001$, $^{****}p < 0.0001$, ns = not significant.

decrease compared with the Vehicle ($F_{1, 14} = 2.48$, $p < 0.01$) and the Cur-Con ($F_{1, 14} = 0.01$, $p < 0.0001$) groups (Fig 3A). As expected, compared to the D-gal treated mice group, a significant protection from the decreasing tendency of FR was detected in the Curcumin + D-gal ($62.18 \pm 6.57\%$; $F_{1, 14} = 5.44$, $p < 0.001$) mice group (Fig 3A). This FR was comparable to the FR exhibited by the Ast (standard antioxidant) + D-gal mice ($63.95 \pm 3.95\%$; $F_{1, 14} = 2.54$, $p > 0.05$; Fig 3A). A similar pattern was observed in the NA group. During the last 6 min with CS-US pairings, the NA group displayed an FR of $35.83 \pm 0.62\%$ (Fig 3B). A significant protection from the decreasing tendency of FR was found in Curcumin + NA ($58.95 \pm 3.16\%$; $F_{1, 14} = 6.89$, $p < 0.0001$; Fig 3B) group mice (Fig 3B). An equivalent FR elevation was also apparent in the Ast (standard antioxidant drug) + NA group ($59.37 \pm 2.13\%$; $F_{1, 14} = 1.16$, $p > 0.05$; Fig 3B).

**3.2.2. Effects of curcumin on freezing response in context test.** During the last 6 min with CS-US pairings, the Vehicle and Cur-Con groups exhibited FR of $64.72 \pm 0.70\%$ and $73.05 \pm 0.74\%$, respectively. The FR was $52.77 \pm 1.85\%$ in the D-gal treated mice group, indicating a significant decrease compared with the Vehicle ($F_{1, 14} = 4.83$, $p < 0.001$) and Cur-Con ($F_{1, 14} = 4.63$, $p < 0.0001$) groups (Fig 4A). On the contrary, a significant protection from the declining trend of FR was detected in the Curcumin + D-gal ($62.01 \pm 0.83\%$; $F_{1, 14} = 3.38$, $p < 0.05$) and was comparable to the FR exhibited by the Ast (standard antioxidant drug) + D-gal treated mice group ($61.59 \pm 2.88\%$; $F_{1, 14} = 9.30$, $p > 0.05$; Fig 4A). During the last 6 min with CS-US pairings, the NA group displayed an FR of $50.13 \pm 0.57\%$ (Fig 4B). This decreasing tendency of FR was remarkably prevented in the Curcumin + NA mice group ($60.27 \pm 1.38\%$; $F_{1, 14} = 9.72$, $p < 0.01$; Fig 4B). This FR was comparable to the FR exhibited by the standard drug (Ast) in Ast + NA ($62.36 \pm 3.16\%$; $F_{1, 14} = 7.81$, $p > 0.05$; Fig 4B) mice group.

**3.2.3. Effects of curcumin on freezing response in cued test.** In the cued test following the context test, mice were placed in a differently shaped chamber with altered context. During the first 3 min period with Pre-CS presentation, the baseline activity was almost similar among Vehicle, Cur-Con, D-gal, Curcumin + D-gal, and Ast (standard antioxidant) + D-gal groups (Fig 4A). On day 2(2b), during the last 3 min of the cued test with CS presentation, the Vehicle

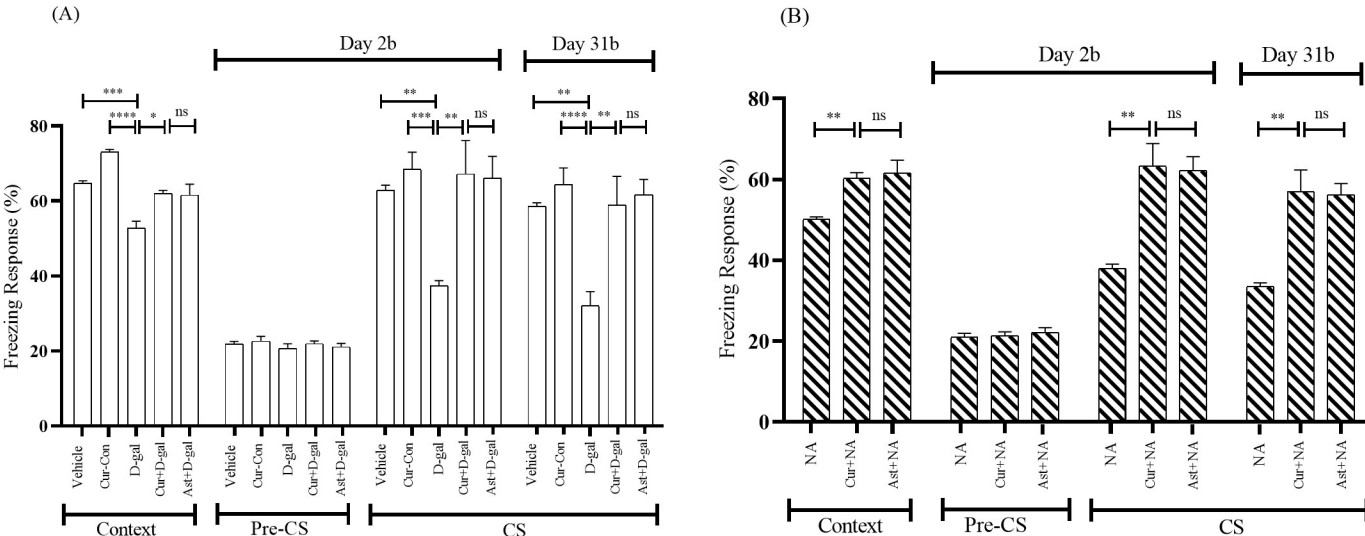

**Fig 4.** Effect of curcumin on the context and cued fear memory (Day 2b and 31b) of (A) D-gal and (B) NA mice group. The memory was assessed by analyzing the FR. The FR was expressed in percentage (%). Data was presented as mean ± SEM, n = 8 each group; *$p < 0.05$, **$p < 0.01$, ***$p < 0.001$, ****$p < 0.0001$, ns = not significant.

and Cur-Con groups exhibited an FR of 62.84 ± 1.37% and 68.43 ± 4.58%, respectively (Fig 4A). The percentage of FR was considerably low in the D-gal administered mice (37.43 ± 1.36%), significantly high both in the Vehicle ($F_{1, 14} = 0.30$, $p < 0.01$) and and Cur-Con ($F_{1, 14} = 4.62$, $p < 0.001$). Compared to the decreasing tendency of the percentage of FR displayed by the D-gal treated mice group, there was significant protection detected in the Curcumin + D-gal (67.23 ± 8.87%) and Ast + D-gal mice groups (66.11 ± 5.82%) (Fig 4A). The differences in FR were remarkable between D-gal and Curcumin + D-gal groups ($F_{1, 14} = 26.64$, $p < 0.01$) but indistinguishable between Curcumin + D-gal and standard drug (Ast) + D-gal groups ($F_{1, 14} = 3.15$, $p > 0.05$; Fig 4A). The baseline activity did not differ among the NA, Curcumin + NA, and Ast + NA groups (Fig 4B). On day 2(2b), during the last 3 min of the cued test with CS presentation, the NA group displayed an FR of 38.05 ± 1.03% (Fig 4B). On the contrary, the declining tendency of FR was prevented in the Curcumin + NA mice group (63.28 ± 5.58%; Fig 4B). One-Way ANOVA revealed that the FR elevation in Curcumin + NA mice was statistically significant ($F_{1, 14} = 13.69$, $p < 0.01$) compared to the NA mice and comparable to that in the standard antioxidant group (Ast + NA) mice group (62.22 ± 3.39%; $F_{1, 14} = 2.51$, $p > 0.05$; Fig 4B).

On day 31(31b), during the last 3 min of the cued test with CS presentation, The percentage of FR was 32.15 ± 3.73 in the D-gal injected mice group, indicating a significant decrease compared with the Vehicle (58.54 ± 0.98%; $F_{1, 14} = 7.94$, $p < 0.01$) and the Cur-Con (64.47 ± 4.36%; $F_{1, 14} = 0.34$, $p < 0.0001$) groups (Fig 4A). Compared to the declining trend of the percentage of FR displayed by the D-gal treated mice group, significant protection was seen in Curcumin + D-gal treated mice group (58.98 ± 7.58%; $F_{1, 14} = 6.09$, $p < 0.01$; Fig 4A). This percentage of FR was similar to the FR exhibited by the standard drug (Ast) + D-gal-treated mice (61.66 ± 4.09%; $F_{1, 14} = 4.89$, $p > 0.05$; Fig 4A). A similar trend was observed in the NA group. During the last 3 min of the cued test with CS presentation, the NA group exhibited an FR of 33.47 ± 0.90% (Fig 4B). Contrarily, this change of FR was remarkably protected in the Curcumin + NA mice group (57.03 ± 5.29%; $F_{1, 14} = 8.47$, $p < 0.01$; Fig 4B). This FR was comparable to the FR exhibited by the Ast + NA group (56.18 ± 2.81%; $F_{1, 14} = 2.34$, $p > 0.05$; Fig 4B).

### 3.3. Effects of curcumin on oxidative stress biomarkers

**3.3.1. Glutathione.** The GSH level detected in the Vehicle and Cur-Con groups were 9.34 ± 0.87 μmol/mg and 12.72 ± 0.97 μmol/mg, respectively (Fig 5A). The level of GSH in the D-gal treated mice group was 3.19 ± 0.31 μmol/mg, illustrating a significant decrease of GSH compared with Vehicle ($F_{1, 14}$ = 3.63, $p < 0.0001$) and Cur-Con ($F_{1, 14}$ = 2.58, $p < 0.0001$) groups (Fig 5A). The GSH level in Curcumin + D-gal mice group was 10.77 ± 1.02 μmol/mg, indicating a remarkable protection from the decreasing trend of GSH level ($F_{1, 14}$ = 16.49, $p < 0.0001$) (Fig 5A). A similar GSH elevation was apparent in the Ast + D-gal mice (10.81 ± 0.52 μmol/mg; $F_{1, 14}$ = 7.73, $p > 0.05$; Fig 5A).

The GSH level in the NA group was 3.78 ± 0.59 μmol/mg (Fig 5B). Conversely, this decreasing trend of GSH level was remarkably prevented in the Curcumin + NA mice group (10.16 ± 0.74 μmol/mg; $F_{1, 14}$ = 0.46, $p < 0.0001$; Fig 5B). This GSH level was comparable to the GSH level detected in the Ast + NA (11.28 ± 0.61 μmol/mg; $F_{1, 14}$ = 0.74, $P > 0.05$; Fig 5B) mice group.

**3.3.2. Superoxide dismutase.** The SOD activity was 27.16 ± 0.32 U/30s and 40.24 ± 3.46 U/30s in the Vehicle and Cur-Con groups, respectively (Fig 5C). The SOD activity in D-gal administered mice group was 11.45 ± 0.27 U/30s, indicating a significant decrease in the activity compared with Vehicle ($F_{1, 14}$ = 0.25, $p < 0.05$) and Cur-Con ($F_{1, 14}$ = 21.52, $p < 0.0001$) groups (Fig 5C). Interestingly, curcumin produced an efficient protection of the SOD activity (35.68 ± 3.46 U/30s) detected in Curcumin + D-gal mice group, which was statistically significant ($F_{1, 14}$ = 28.37, $p < 0.0001$) compared to the D-gal treated mice group but insignificant compared to the Ast + D-gal group (37.27 ± 5.46 U/30s; $F_{1, 14}$ = 0.67, $p > 0.05$; Fig 5C) mice.

The SOD activity in the NA mice group was 13.03 ± 0.26 U/30s (Fig 5D). Contrarily, this decreasing activity was protected in the Curcumin + NA mice group (31.27 ± 1.63 U/30s; Fig 5D). A One-Way ANOVA followed by the Brown-Forsythe test demonstrated a statistically significant change in the SOD activity of the curcumin mice compared to the SOD activity of the NA mice ($F_{1, 14}$ = 7.15, $p < 0.01$). The effect of curcumin on the SOD activity was very similar to the effect of Ast (35.39 ± 4.80 U/30s; $F_{1, 14}$ = 14.68, $p > 0.05$; Fig 5D) found in Ast + NA group mice.

**3.3.3. Catalase.** The CAT activity was 6.83 ± 0.39 μmol/min/mg and 12.33 ± 0.92 μmol/min/mg in the Vehicle and Cur-Con groups, respectively. The CAT activity in D-gal treated mice group was 2.82 ± 0.18 μmol/min/mg, illustrating a significant decrease in activity

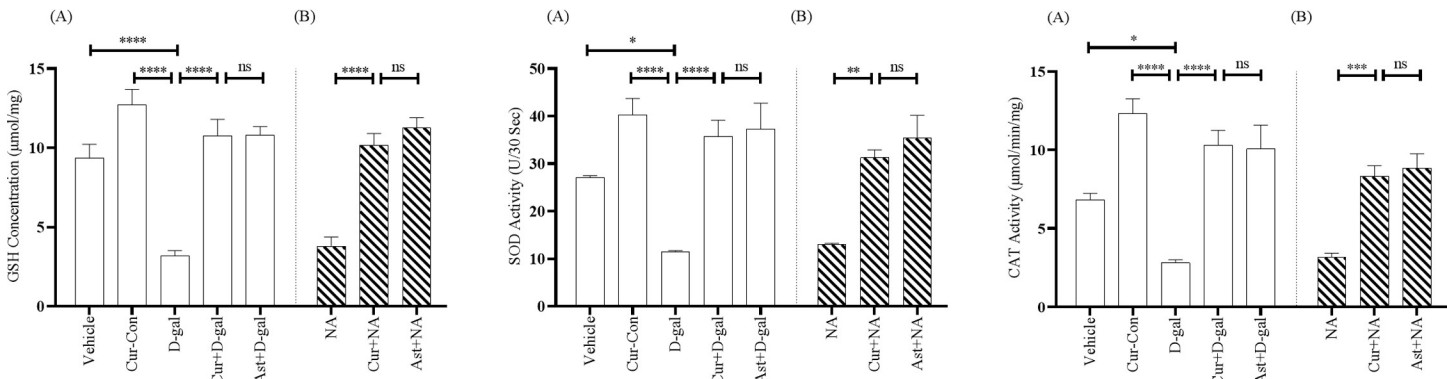

**Fig 5.** Effect of curcumin on GSH concentration (A and B), SOD (C and D) and CAT (E and F) activity in D-gal and NA mice group. The GSH, SOD, CAT was detected by using bioassay technique among Vehicle, Cur-Con, D-gal, Curcumin + D-gal, Ast + D-gal, NA, Curcumin + NA, Ast + NA groups. GSH level, SOD and CAT activity were expressed in μmol/mg, U/30s, and μmol/min/mg, respectively. Data was presented as mean ± SEM, n = 8 each group; *$p < 0.05$, **$p < 0.01$ ***$p < 0.001$, ****$p < 0.0001$ ns = not significant.

compared with Vehicle ($F_{1, 14}$ = 0.81, p < 0.05) and Cur-Con ($F_{1, 14}$ = 29.53, p < 0.0001) groups (Fig 5E). On the contrary, a statistical significant protection from the decreasing activity of CAT activity was detected in the Curcumin + D-gal mice group (10.29 ± 0.95 µmol/min/mg; $F_{1, 14}$ = 4.6, p < 0.0001; Fig 5E). This activity was comparable to the CAT activity of Ast + D-gal mice group (10.06 ± 1.51 µmol/min/mg; $F_{1, 14}$ = 0.78, p > 0.05; Fig 5E).

Similarly, the CAT activity in the NA mice was 3.15 ± 0.26 µmol/min/mg (Fig 5F). Curcumin produced efficient protection of the CAT activity (8.31 ± 0.22 µmol/min/mg Fig 5F), detected in the Curcumin + NA mice group. A One-Way ANOVA followed by the Brown-Forsythe test confirmed a statistically significant change in the CAT activity by curcumin ($F_{1, 14}$ = 0.47, p < 0.001) compared to the CAT activity of the NA mice. This CAT activity was comparable to that of the Ast + NA mice (8.85 ± 0.91 µmol/min/mg; $F_{1, 14}$ = 3.24, p > 0.05; Fig 5F).

**3.3.4. Advanced oxidation of protein products.** The AOPP level was drastically elevated to 146.41 ± 16.60 µmol/ml in the D-gal group, illustrating a significant increase of AOPP level compared with the Vehicle (84.80 ± 11.87 µmol/ml; $F_{1, 14}$ = 0.99, p < 0.01) and Cur-Con (49.65 ± 7.03 µmol/ml; $F_{1, 14}$ = 5.11, p < 0.0001) groups (Fig 6A). However, a statistical significant protection from the increasing tendency of AOPP level was found in the Curcumin + D-gal mice group (61.55 ± 10.80 µmol/ml; Fig 6A; $F_{1, 14}$ = 2.30, p < 0.0001), but was indistinguishable from the Ast + D-gal group (56.44 ± 10.44 µmol/ml; $F_{1, 14}$ = 0.13, p > 0.05; Fig 6A). Similarly, the AOPP level was significantly declined to 69.52 ± 9.90 µmol/ml in the Curcumin + NA mice group compared to the NA (137.67 ± 15.24 µmol/ml) mice ($F_{1, 14}$ = 0.76, p < 0.01; Fig 6B) group. This level was comparable to the level of Ast + NA mice group (61.28 ± 7.04 µmol/ml; $F_{1, 14}$ = 0.28, p > 0.05; Fig 6B).

**3.3.5. Nitric oxide.** D-gal administration profoundly enhanced the NO level to 9.40 ± 1.01 mmol/mg, indicating a significant increase of NO level compared with the Vehicle (4.62 ± 0.33 mmol/mg; $F_{1, 14}$ = 6.02, p < 0.0001) and Cur-Con (3.23 ± 0.75 mmol/mg; $F_{1, 14}$ = 0.22, p < 0.0001) groups (Fig 6C). Conversely, a statistical significant protection from the increasing tendency of NO level was detected in the Curcumin + D-gal mice group (4.47 ± 0.53 mmol/mg; $F_{1, 14}$ = 2.75, p < 0.0001; Fig 6C). This level was comparable to the NO level of Ast + D-gal mice (4.09 ± 0.55 mmol/mg; $F_{1, 14}$ = 0.32, p > 0.05; Fig 6C). A similar pattern was observed in the normal-aged mice. The NO level in the NA group was 8.06 ± 0.29 mmol/mg (Fig 6D). On the contrary, this increasing tendency of NO level was prevented in the Curcumin + NA (5.28 ± 0.25 mmol/mg) group mice (Fig 6D). A One-Way ANOVA followed by the Brown-Forsythe test demonstrated a statistically significant effect of curcumin

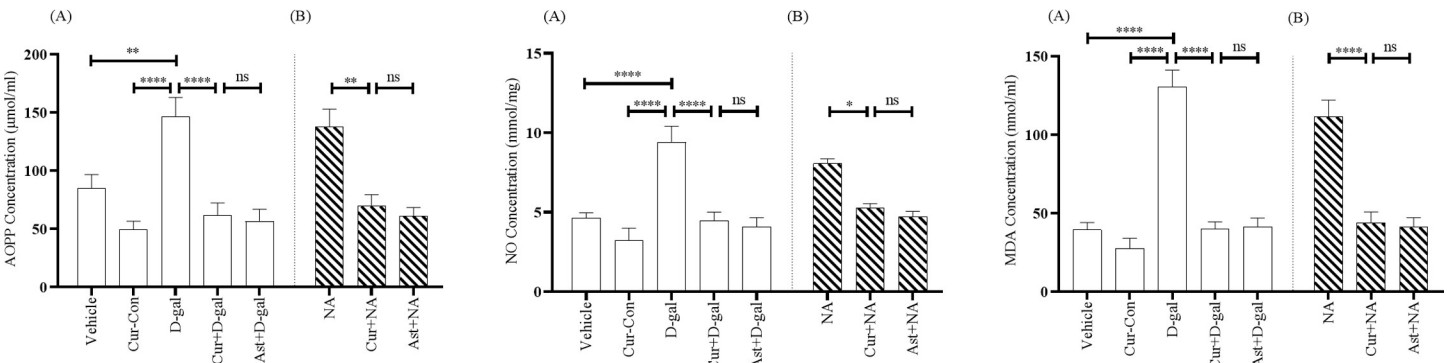

**Fig 6.** Effect of curcumin on AOPP (A and B), NO (C and D), and MDA (E and F) concentration in D-gal and NA mice group. The AOPP level was assessed using bioassay technique among Vehicle, Cur-Con, D-gal, Curcumin + D-gal, Ast + D-gal, NA, Curcumin + NA, Ast + NA groups. AOPP, NO, and MDA level was represented in µmol/ml, mmol/mg, and nmol/ml, respectively. Data was presented as mean ± SEM, n = 8 each group; *p < 0.05, **p < 0.01, ****p < 0.0001 ns = not significant.

**Table 2. Estimated binding energies from Vina molecular docking.**

| Protein | PDB ID | Reference Ligand | Binding Energy (kcal/mol) | |
| --- | --- | --- | --- | --- |
| | | | Curcumin | Reference Ligand |
| BACE1 | 2QP8 | SCH734723 | -8.5 | 10 |
| GSTA1 | 4HJ2 | Chlorambucil | -9.6 | -7.5 |
| GSTO1 | 4YQM | C1-27 | -8.6 | -6.6 |
| KEAP1 | 6TYM | 08A | -8.6 | -9.2 |
| MAOA | 2Z5X | Harmine | -9.2 | -8.7 |

($F_{1, 14}$ = 0.91, p < 0.05). The NO level in the curcumin mice was comparable to that of the Ast + NA mice group (4.72 ± 0.33 mmol/mg; $F_{1, 14}$ = 1.64, p > 0.05; Fig 6D).

**3.3.6. Lipid peroxidation.** The MDA level was 39.37 ± 4.62 nmol/ml and 27.64 ± 6.40 nmol/ml in the Vehicle and Cur-Con groups, respectively. The level of MDA in the D-gal group was 130.37 ± 10.81 nmol/ml, indicating a significant increase in the MDA level compared with Vehicle ($F_{1, 14}$ = 5.32, p < 0.0001) and Cur-Con ($F_{1, 14}$ = 2.69, p < 0.0001) groups (Fig 6E). A statistical significant protection from the increasing trend of MDA level was found in the Curcumin + D-gal mice group (40.05 ± 4.39 nmol/ml; $F_{1, 14}$ = 5.66, p < 0.0001; Fig 6E). This MDA level was comparable to that of the Ast + D-gal mice group (41.29 ± 5.53 nmol/ml; $F_{1, 14}$ = 0.78, p > 0.05; Fig 6E). Similarly, the MDA level in the NA group was 111.50 ± 10.49 nmol/ml (Fig 6F). Interestingly, this increasing pattern of the MDA level was significantly prevented in the Curcumin + NA mice group (43.74 ± 6.97 nmol/ml; $F_{1, 14}$ = 0.53, p < 0.0001) and was comparable to the Ast + NA mice group (41.08 ± 6.06 nmol/ml; ($F_{1, 14}$ = 0.31, p > 0.05; Fig 6F).

## 3.4. Molecular docking

Curcumin significantly attenuated oxidative stress in our in vivo aging model. We performed molecular docking to predict interactions of curcumin with glutathione S-transferase A1 (GSTA1), glutathione S-transferase omega-1(GSTO1), and kelch-like ECH-associated protein 1(KEAP1), which play significant roles in redox signaling to regulate cellular events such as senescence. GSTA1 and GSTO1, which belong to the ROS/RNS neutralizing enzyme gene family glutathione transferase (GST), catalyze GSH conjugation with reactive electrophiles and detoxify hydroperoxides for maintaining redox homeostasis [50]. KEAP1, a central key sensor that regulates the expression of many cytoprotective genes in oxidative and electrophile stress, targets nuclear factor-erythroid 2-related factor 2 (Nrf2) in modulating redox homeostasis [51]. We first performed Vina molecular docking simulations of curcumin using 3D crystallographic structures of GSTA1, GSTO1, and KEAP1. Estimated binding energy values (kcal/mol) are given in Table 2. In general, a more negative value is an indication of a more stable complex. When docked to GSTA1, curcumin had lower binding energy than the co-crystallized ligand chlorambucil (-9.6 vs. -7.5) (Table 2). Analysis of docking poses revealed that the chlorambucil was redocked at the same binding pocket with a slightly different binding pose than observed in the pdb file (Fig 7A). The predicted binding site of curcumin matched that of redocked chlorambucil (Fig 7B). In the GSTA1 crystal structure, chlorambucil forms a salt bridge, three hydrogen bonds, and multiple van der Waals interactions (Fig 7C). On the other hand, curcumin is predicted to form a pi-sigma bond with ALA100 and multiple van der Waals interactions (Fig 7D). Both curcumin and chlorambucil can interact with the residues GLY14, THR68, ILE106, LEU107, and MET208 of GSTA1 through van der Waals interactions. Overall, curcumin is predicted to bind GSTA1 with favorable interactions. Compared to the

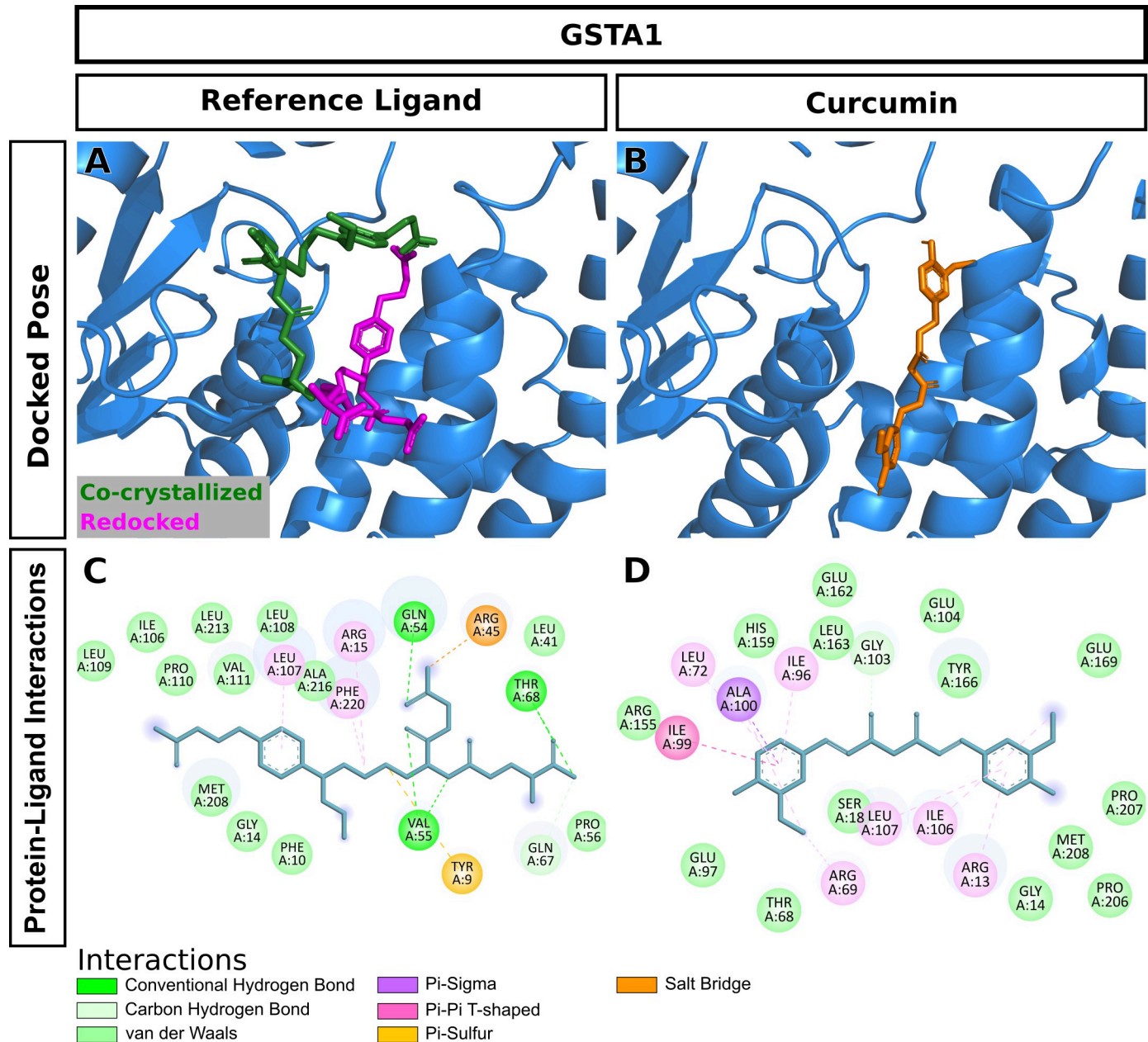

**Fig 7. Molecular docking of curcumin to GSTA1.** (A) Redocking of the bound ligand. (B) Docking pose of curcumin. (C) Interactions of the bound ligand with the protein. (D) Predicted interactions of curcumin with the protein.

reference ligand C1-27, curcumin also had a stronger binding affinity for GSTO1 (Table 2). Vina performed exceptionally well to reproduce the docking pose of the co-crystallized inhibitor C1-27 (Fig 8A). Interestingly, curcumin also occupied the binding pocket of GSTO1 (Fig 8B) with an extended structure. More importantly, many interacting residues, including CYS32, PRO33, PHE34, LEU56, VAL72, PRO124, ILE131, TRP180, and TYR229, were common for curcumin and C1-27 (Fig 8C and 8D). It can be assumed that curcumin will be a potent inhibitor for GSTO1. For KEAP1, Vina docking showed a slightly weaker interaction of curcumin than the bound ligand (Table 2). Both curcumin and the reference ligand 08A were

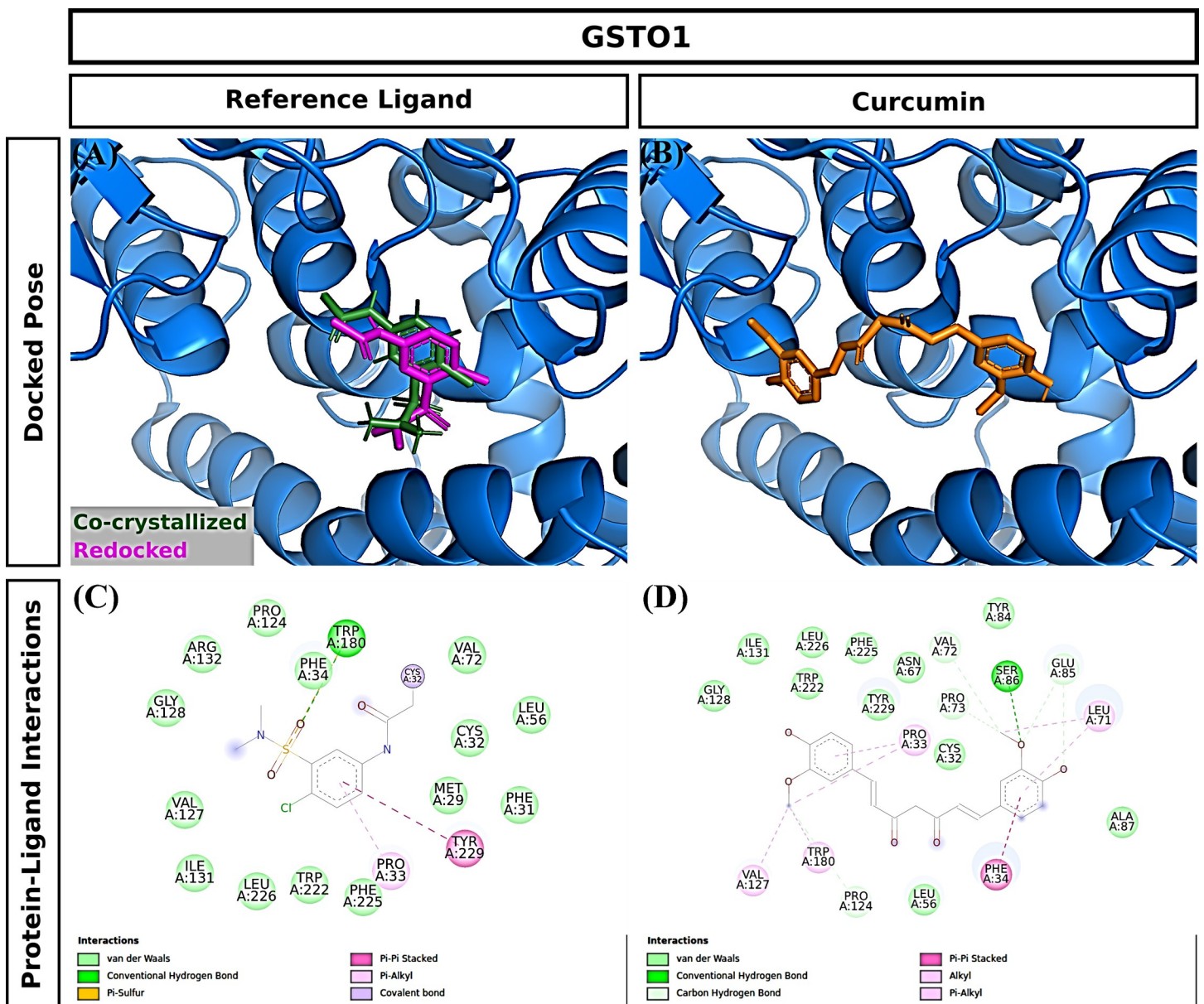

**Fig 8. Molecular docking of curcumin to GSTO1.** (A) Redocking of the bound ligand. (B) Docking pose of curcumin. (C) Interactions of the bound ligand with the protein. (D) Predicted interactions of curcumin with the protein.

docked at the binding pocket of KEAP1 with a different pose than that of the bound ligand in the crystal structure (Fig 9A and 9B). Many residues interacting with 08A were also predicted to form van der Waals interactions with curcumin (Fig 9C and 9D). Curcumin forms at least two hydrogen bonds and one Pi-Sigma bond with KEAP1 (Fig 9D). Thus, curcumin is predicted as a ligand for KEAP1. To explore additional molecular targets of Cur, we used the SwissTargetPrediction (http://www.swisstargetprediction.ch) webserver. Among the top 50 targets, 26% (15/50) were kinases, 16% lyases (8/50), 10% (5/50) proteases, and 6% (3/50) oxidoreductases (Fig 10). The top ten predicted target proteins are shown in Fig 11. The predicted targets also included five proteins, including the oxidoreductase MAOA, with a probability of 1 (on a scale of 0–1), for which curcumin is a known active. Next to these perfect hits, BACE1 yielded

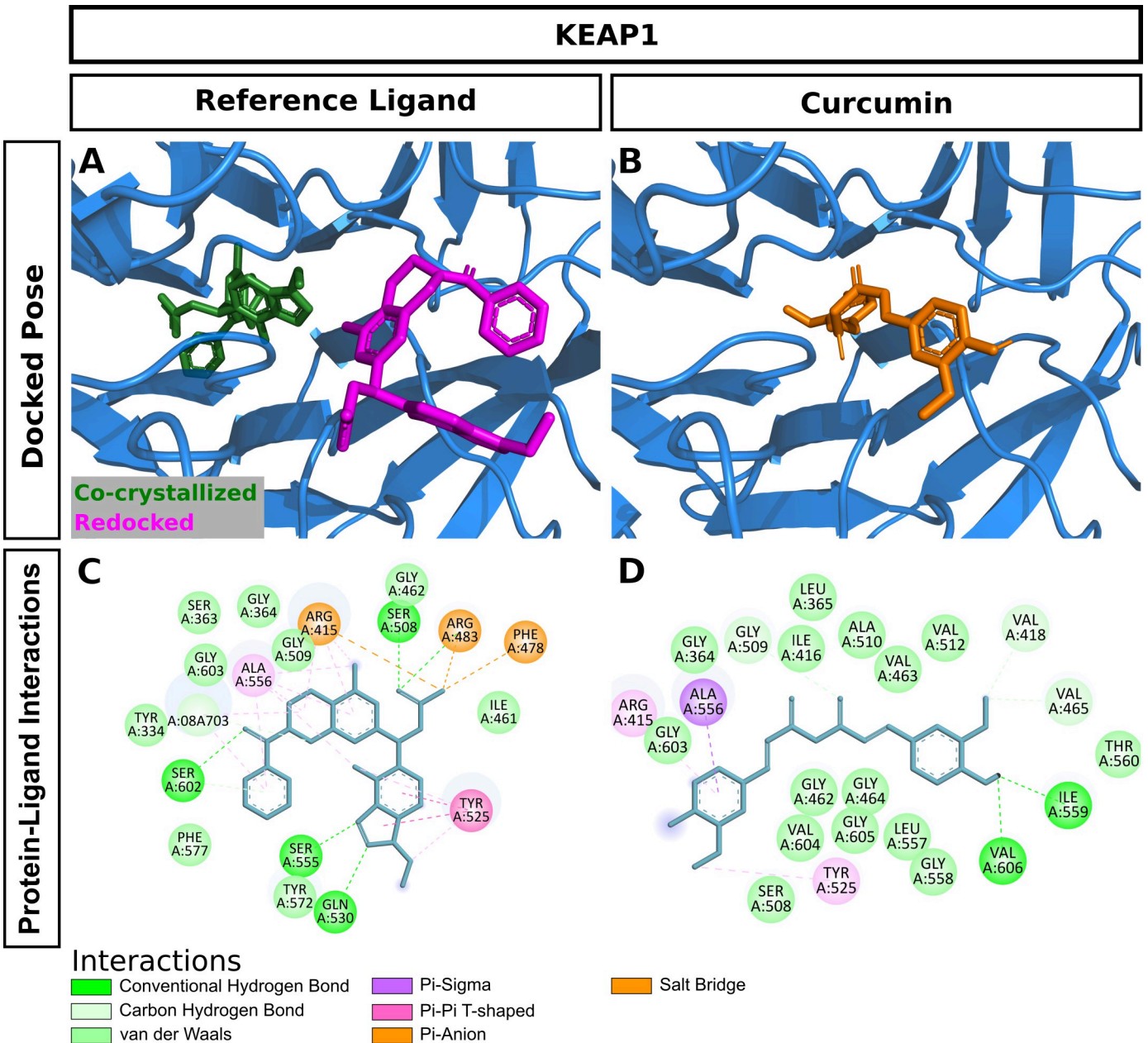

**Fig 9. Molecular docking of curcumin to KEAP1.** (A) Redocking of the bound ligand. (B) Docking pose of curcumin. (C) Interactions of the bound ligand with the protein. (D) Predicted interactions of curcumin with the protein.

a very high probability score (0.83) (Fig 11). Vina docking showed a higher binding affinity (Table 2) and comparable binding interactions of curcumin (Fig 12B and 12D) and the bound ligand harmine (Fig 12A and 12C) for MAOA, supporting the SwissTargetPrediction results. For BACE1, the estimated binding energies were -8.5 and -10 for curcumin and SCH734723, respectively (Table 2). In this case, Vina redocking perfectly matched the docking pose of SCH734723 in the BACE1 crystal structure (Fig 13A). Curcumin was also predicted to dock at the same binding pocket (Fig 13B). Intriguingly, multiple similar amino acid residues of BACE1 are likely to interact with both the reference ligand and curcumin (Fig 13C and 13D).

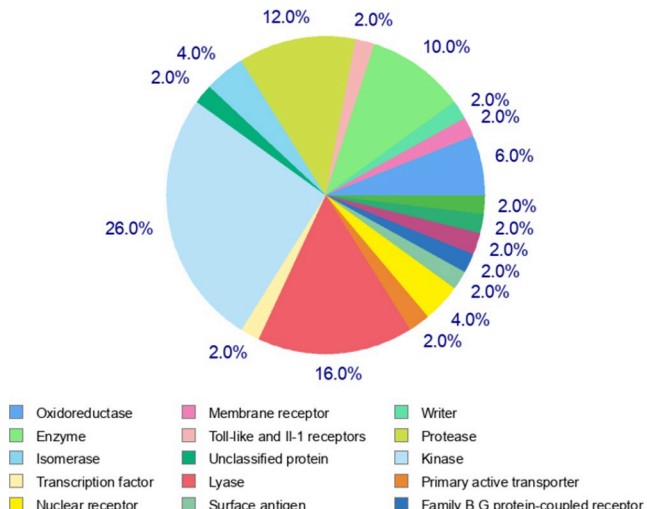

**Fig 10. Relative abundance of the class of top 50 predicted molecular targets of curcumin obtained from SwissTargetPrediction.**

For instance, GLY72, GLN73, ILE171, and TYR132 interact with both molecules. Ser71, GLY74, and ARG296 can form conventional hydrogen bonds with the curcumin. Overall, our molecular docking identifies curcumin as a candidate BACE1 inhibitor.

## 4. Discussion

This current study investigated the effects of curcumin on D-gal and normal aging-induced memory impairment. In vivo study revealed that curcumin protected the decreasing tendency of D-gal and Normal aging-induced RT and FR in PA and CFC tasks. In addition, curcumin ameliorated the level of oxidative stress biomarkers (GSH, SOD, CAT, AOPP, NO MDA). In silico study discerned that curcumin-mediated antioxidant effects in mice could result from, at least partially, binding with several regulatory proteins such as GSTA1, GSTO1, KEAP1, BACE1, and MAOA.

| Target | Common name | Uniprot ID | ChEMBL ID | Target Class | Probability* |
|---|---|---|---|---|---|
| Monoamine oxidase A | MAOA | P21397 | CHEMBL1951 | Oxidoreductase | |
| Beta amyloid A4 protein | APP | P05067 | CHEMBL2487 | Membrane receptor | |
| Histone acetyltransferase p300 | EP300 | Q09472 | CHEMBL3784 | Writer | |
| Prostaglandin E synthase | PTGES | O14684 | CHEMBL5658 | Enzyme | |
| Toll-like receptor (TLR7/TLR9) | TLR9 | Q9NR96 | CHEMBL5804 | Toll-like and Il-1 receptors | |
| Beta-secretase 1 | BACE1 | P56817 | CHEMBL4822 | Protease | |
| DNA topoisomerase II alpha | TOP2A | P11388 | CHEMBL1806 | Isomerase | |
| Nuclear factor erythroid 2-related factor 2 | NFE2L2 | Q16236 | CHEMBL1075094 | Unclassified protein | |
| Arachidonate 5-lipoxygenase | ALOX5 | P09917 | CHEMBL215 | Oxidoreductase | |
| Cyclooxygenase-1 | PTGS1 | P23219 | CHEMBL221 | Oxidoreductase | |

**Fig 11. Names and target probabilities of top 10 predicted molecular targets of curcumin obtained from Swiss TargetPrediction.**

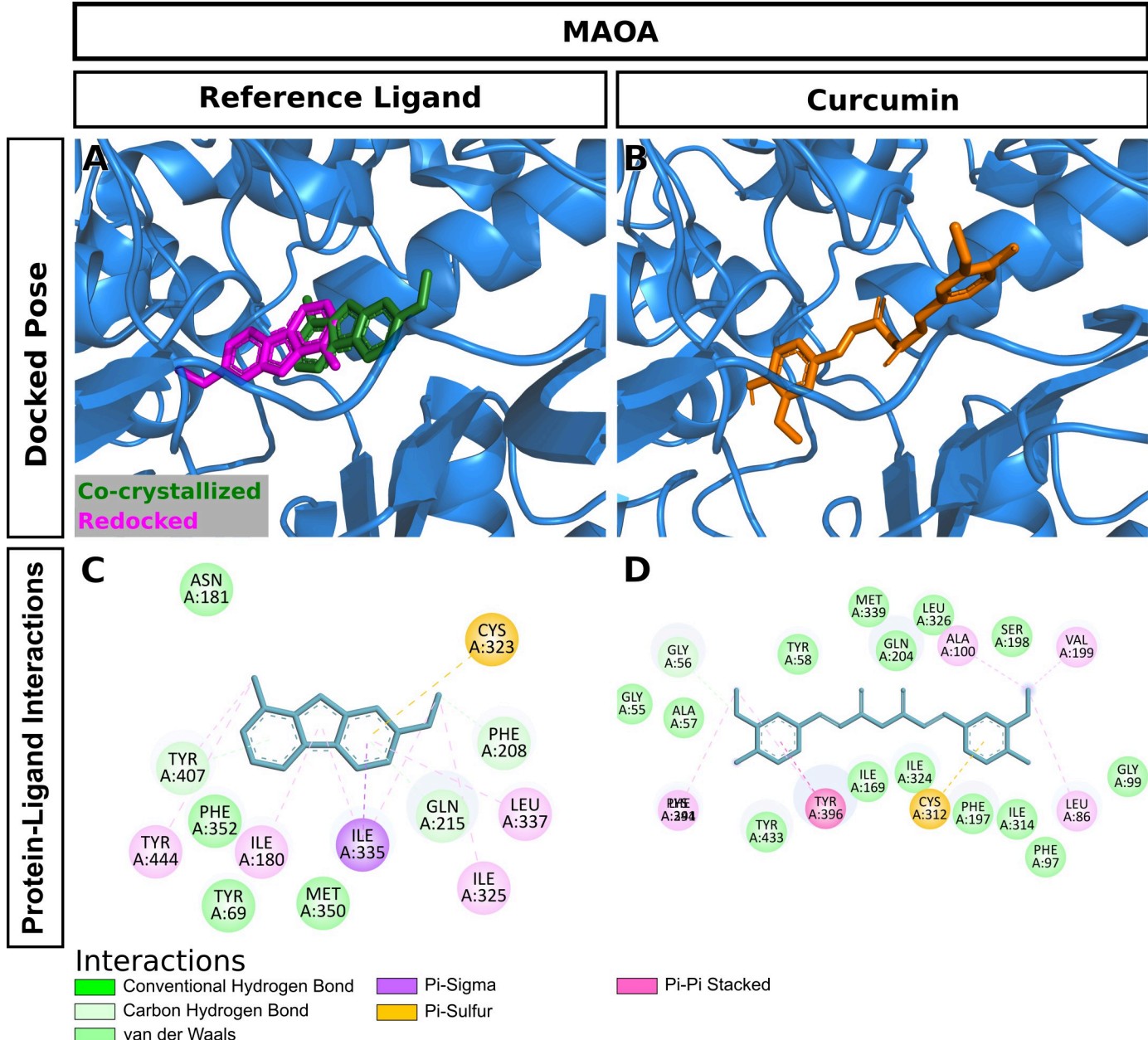

**Fig 12. Molecular docking of curcumin to MAOA.** (A) Redocking of the bound ligand. (B) Docking pose of curcumin. (C) Interactions of the bound ligand with the protein. (D) Predicted interactions of curcumin with the protein.

### 4.1. D-gal facilitates memory impairment and mimics normal-aging in mice

In the present study, D-gal and NA mice groups exhibited less RT and FR in PA and CFC tasks, respectively (Figs 2–4). However, the RT and FR were comparable between D-gal and NA mice groups (Figs 2–4). Several studies revealed that a high dose of D-gal impairs ATP production, redox homeostasis, and increases NADPH which generates excess reactive oxygen species [10]. Other studies showed that a high ROS level could induce neuroinflammation, cellular apoptosis, and brain-derived neurotrophic factor (BDNF) dysregulation [52],

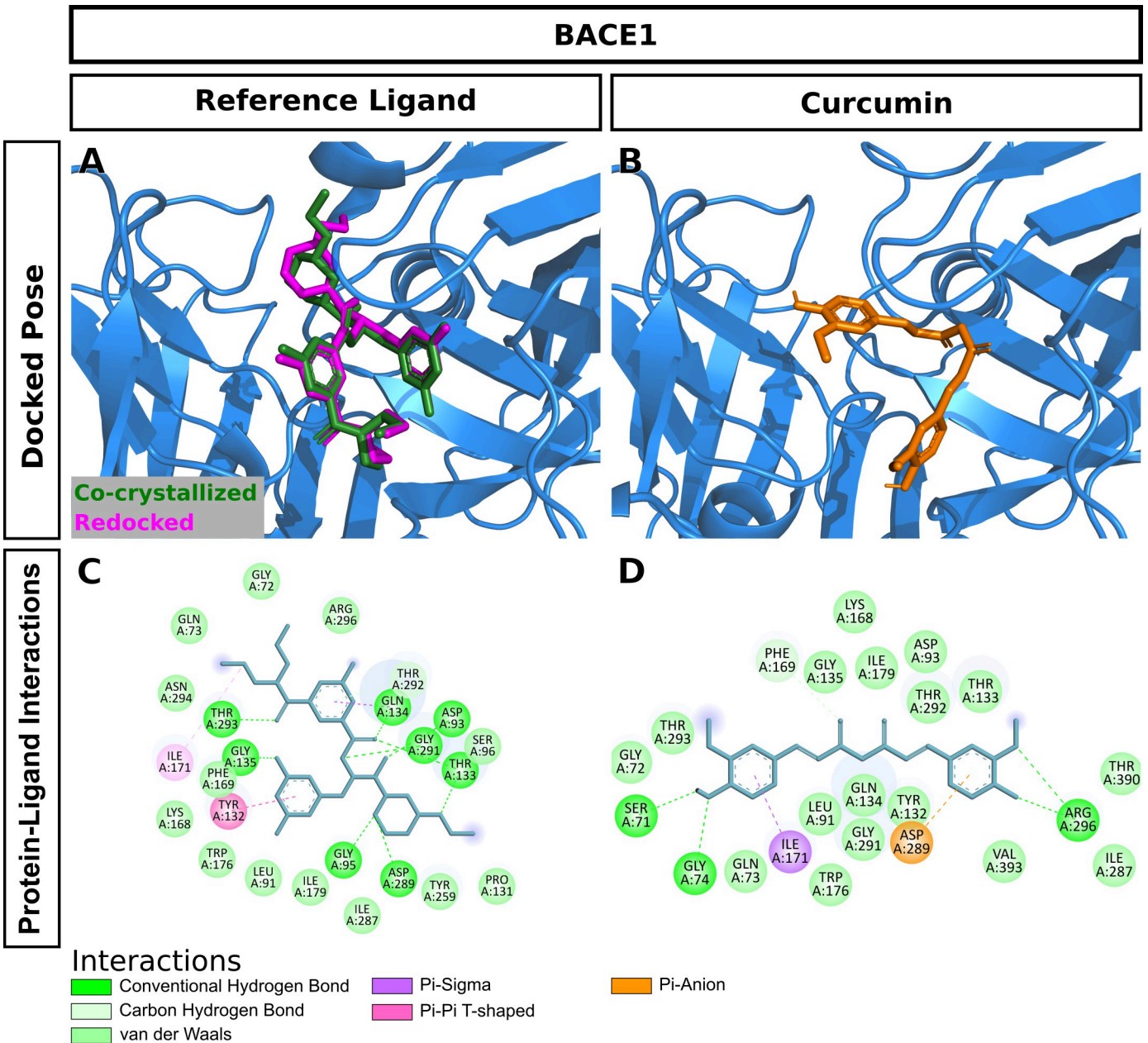

**Fig 13. Molecular docking of curcumin to BACE1.** (A) Redocking of the bound ligand. (B) Docking pose of curcumin. (C) Interactions of the bound ligand with the protein. (D) Predicted interactions of curcumin with the protein.

contributing to cognitive impairment. Another study showed that D-gal reduced the number of new neurons at the subgranular zone in the hippocampus area, worsening memory and the learning process in mice [53]. In addition, D-gal diminishes the expression of synaptic proteins in the hippocampus and the cerebral cortex [54]. Interestingly, D-gal-induced molecular and cellular alterations of the brain observed in experimental animals also appear during the natural progression of aging [55]. Therefore, it can be stated that D-gal mimics natural aging and draws memory deterioration in mice.

## 4.2. Curcumin improves D-gal and normal-aging induced memory impairment

In the current study, Curcumin + D-gal, and Curcumin + NA mice groups showed substantial protection of RT and FR in PA and CFC, respectively (Figs 2–4). The RT and FR were comparable between Curcumin + D-gal and Curcumin + NA mice groups (Figs 2–4). Moreover, curcumin was comparable to Ast, a standard antioxidant, in improving retention and freezing memory. Many studies showed that curcumin possesses anti-neuroinflammation, antioxidant and anti-aging properties [56, 57]. Another study reported that curcumin plays a protective role in brain aging by modulating cell proliferation, neuronal degeneration, and cellular senescence [14]. Therefore, it can be inferred that curcumin improves memory owing to its antioxidant and anti-aging properties in mice. Similar to previous findings [58], our study suggested that curcumin ameliorates memory impairment induced by D-gal or normal-aging in mice.

## 4.3. Curcumin ameliorates the oxidative biomarkers in D-gal and NA mice

In the present study, we found decreased levels of antioxidants such as GSH, SOD, and CAT (Fig 5) in D-gal and NA mice. Moreover, these levels were comparable between D-gal and NA mice (Fig 5). Many studies showed that the abnormal changes of ROS and RNS augment aging processes induced by D-gal [59] and NA [60]. In normal conditions, the antioxidant defense system comprises endogenous non-enzymatic and enzymatic compounds that counteract the deleterious effect of ROS and RNS [61]. Glutathione (GSH), a non-enzymatic antioxidant tripeptide, has a thiol group that interacts with ROS and RNS directly and reduces $H_2O_2$ to form $H_2O$ [62]. In contrast, enzymatic antioxidant Superoxide Dismutase (SOD) catalyzes the dismutation of $O_2^{\bullet-}$ into $H_2O_2$ and $O_2$ [63]. Another enzymatic antioxidant, Catalase (CAT), converts $H_2O_2$ into $H_2O$ and $O_2$; thus, cell protection from deleterious effects of $H_2O_2$ continues [63]. These antioxidants (GSH, SOD, CAT) prevent lipid peroxidation in the cell membrane and maintain redox homeostasis [64]. On the other hand, low levels of GSH, SOD, and CAT fail to protect against the overproduction of ROS and RNS and expedite the aging process [65].

We found a high level of AOPP, NO, and MDA in D-gal and NA mice (Fig 6), suggesting an induction of the aging process [29]. A high level of protein oxidation is known to generate excess AOPP [66]. Nitric Oxide (NO) reacts with superoxide ($O^{2\bullet-}$) to form a stronger oxidant peroxynitrite anion (ONOO⁻). A high level of NO generates more RNS, such as $^{\bullet}NO_2$ and $N_2O_3$ [67], whereas an increased level of MDA promotes ROS generation [68]. Therefore, high levels of MDA, AOPP, and NO contribute to the aging process by increasing oxidative stress in the brain.

We found that both curcumin and Ast prevented the decreasing trend of GSH, SOD, and CAT levels detected in D-gal and NA groups (Fig 5), suggesting that curcumin's antioxidant activity minimized the aging-associated oxidative burden in mice [69]. Our results are also in agreement with previous studies [70].

On the other hand, substantial protection from the increasing tendency of AOPP, NO, and MDA was detected in Curcumin + D-gall and Curcumin +NA mice groups (Fig 6). These effects were comparable in the Ast group (Fig 6). Another study [71] supported these findings, suggesting that curcumin exerts its antioxidant activity by controlling the overproduction of AOPP, NO, and MDA.

### 4.4. Predicted interactions of curcumin with glutathione S-transferase A1, glutathione S-transferase omega-1, kelch-like ECH-associated protein 1, beta-secretase 1, and amine oxidase (flavin-containing) A to exert antioxidant activity

In our in silico studies, estimated binding energies were assessed by adopting Vina molecular docking. Estimated binding energies of curcumin with GSTA1, GSTO1, and KEAP1 were comparable with reference ligands such as Chlorambucil, C1-27, 08A, respectively (Table 2). Furthermore, curcumin is predicted to bind more strongly with GSTA1 and GSTO1 compared with the reference ligands (Table 2).

GSTA1, GSTO1, and KEAP1 are abundantly present in the hippocampus, a critical brain region crucial for hippocampus-dependent learning tasks [72, 73]. Western blot analysis showed that the upregulation of GSTA1 in the CA1 area [74] and the downregulation of GSTO1 in the hippocampus were linked to cognitive impairment, [75] commonly seen in aging animals. Other studies showed that these proteins are closely associated with oxidative stress and aging-induced neurodegenerative diseases such as memory impairment [76]. GSTA1 and GSTO1 are primary phase II detoxification enzymes and catalyze GSH conjugation in the presence of electrophile substrates [19]. Furthermore, GSTA1 suppresses the activation of c-Jun N-terminal kinase (JNK) signaling by a pro-inflammatory cytokine and oxidative stress [77]. Moreover, GSTO1 regulates the activation of interleukin-1β and stops the inflammation process in aging-associated neurodegenerative disease [20]. The Keap1-Nrf2 system plays a crucial role in regulating oxidative stress-mediated disorders [78]. A western blot analysis found a lower expression of KEAP1 in CA3 and dentate gyrus of the hippocampus under oxidative conditions [78]. The KEAP1 is closely associated with the Nrf2 cytoprotective signaling pathway and plays an antioxidative role. Under the homeostatic state, KEAP1 controls the level of Nrf2 upon binding. During stressful conditions, the KEAP1 gets oxidized in the presence of electrophile, stopping Nrf2 ubiquitylation. These cause Nrf2 to move into the nucleus forming a heterodimer with musculoaponeurotic proteins (Mafs) and initiating cytoprotective molecules such as GSH, SOD, CAT after binding with antioxidant response element (ARE) [21]. Under oxidative stress conditions, the expression of GSTA1, GSTO1, and KEAP1 are down-regulated. In contrast, the expressions are reversed upon treating with curcumin [18, 79], suggesting that curcumin is predicted to interact with GSTA1, GSTO1, and KEAP1 and potentiates the antioxidant activity.

Opposite to the down-regulation of GSTA1, GSTO1, and KEAP1, the BACE1 and MAOA proteins are elevated during the aging-associated memory impairment [80, 81]. Immunohistochemical studies showed a high level of MAOA was found in the CA3 area of the hippocampus, an important region sensitive for brain aging [82]. Studies showed that BACE1 and MAOA proteins are strongly linked to aging-associated memory impairment [22, 83]. BACE1 is widely distributed in CA1 and CA3 areas, and the absence of this protein is responsible for the altered level of synaptic plasticity in aging mice [84]. Furthermore, BACE1 plays a vital role in cleaving Aβ-peptide and causing the accumulation of amyloid-β (Aβ) peptides in the brain [85]. Likewise, MAOA produces hydrogen peroxide by oxidation of monoamine substrates in the mitochondrial outer membrane and facilitates oxidative stress [23]. We found that the estimated binding energies of curcumin upon BACE1 and MAOA were comparable with respective reference ligands (SCH734723, Harmine, respectively) (Table 2), suggesting that curcumin is predicted to interact with BACE1 and MAOA and potentiates its antioxidant role in brain aging.

## 5. Conclusion

We investigated the detailed effects of curcumin on oxidative stress in the D-gal and nature-induced aging mice model. Our in vivo study suggested that curcumin improves memory and rescues learning impairment by modulating oxidative stress levels. Furthermore, our in-silico study demonstrated that curcumin has good binding affinities for several molecular targets implicated in redox homeostasis. Finally, based on our in vivo and computational studies, it can be stated that curcumin improves D-gal and Normal aging-associated memory impairment by reducing oxidative overload in mice.

## Supporting information

**S1 Fig. Effect of curcumin on RT in D-gal and NA mice group after 48 hours of training.** The RT was calculated by performing PA tasks among Vehicle, Cur-Con, D-gal, Curcumin + D-gal, Ast + D-gal, NA, Curcumin + NA, Ast + NA groups. RT was expressed in second. Data was presented as mean ± SEM, n = 8 each group; $^{****}$p $<$ 0.0001, ns = not significant. (TIF)

**S1 Data.**
(XLS)

## Author Contributions

**Conceptualization:** Md. Ashrafur Rahman, Arif Anzum Shuvo, Manik Chandra Shill, Ghazi Mohammad Sayedur Rahman, Hasan Mahmud Reza.

**Data curation:** Md. Ashrafur Rahman, Hasan Mahmud Reza.

**Formal analysis:** Md. Ashrafur Rahman, Arif Anzum Shuvo, Asim Kumar Bepari, Mehedi Hasan Apu, Manik Chandra Shill, Murad Hossain, Monjurul Kader Bakshi, Hasan Mahmud Reza.

**Investigation:** Md. Ashrafur Rahman, Arif Anzum Shuvo, Mehedi Hasan Apu, Murad Hossain, Monjurul Kader Bakshi.

**Methodology:** Md. Ashrafur Rahman, Arif Anzum Shuvo, Asim Kumar Bepari, Monjurul Kader Bakshi, Hasan Mahmud Reza.

**Project administration:** Md. Ashrafur Rahman, Mehedi Hasan Apu, Mohammed Uddin.

**Resources:** Manik Chandra Shill, Murad Hossain, Mohammed Uddin, Md. Rabiul Islam, Ghazi Mohammad Sayedur Rahman.

**Software:** Md. Ashrafur Rahman, Asim Kumar Bepari, Atiqur Rahman.

**Supervision:** Md. Ashrafur Rahman, Manik Chandra Shill, Murad Hossain, Mohammed Uddin, Javed Hasan, Hasan Mahmud Reza.

**Validation:** Md. Ashrafur Rahman, Mehedi Hasan Apu.

**Visualization:** Md. Ashrafur Rahman, Hasan Mahmud Reza.

**Writing – original draft:** Md. Ashrafur Rahman, Arif Anzum Shuvo, Asim Kumar Bepari, Mohammed Uddin, Md. Rabiul Islam, Javed Hasan, Ghazi Mohammad Sayedur Rahman, Hasan Mahmud Reza.

**Writing – review & editing:** Md. Ashrafur Rahman, Asim Kumar Bepari, Md. Rabiul Islam, Atiqur Rahman, Ghazi Mohammad Sayedur Rahman, Hasan Mahmud Reza.

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
