## [Decision Letter · Decision Letter 0]

13 Jan 2022

PONE-D-21-30364Curcumin improves D-galactose and Normal-aging Associated Memory Impairment in Mice: in vivo and in silico-Based StudiesPLOS ONE

Dear Dr. Rahman,

Thank you for submitting your manuscript to PLOS ONE. After careful consideration, we feel that it has merit but does not fully meet PLOS ONE’s publication criteria as it currently stands. Therefore, we invite you to submit a revised version of the manuscript that addresses the points raised during the review process.

We look forward to receiving your revised manuscript.

Kind regards,

Michelle Melgarejo da Rosa

Academic Editor

PLOS ONE

Journal Requirements:

Reviewers' comments:

Reviewer's Responses to Questions

**Comments to the Author**

1. Is the manuscript technically sound, and do the data support the conclusions?

Reviewer #1: No

Reviewer #2: Yes

2. Has the statistical analysis been performed appropriately and rigorously? 

Reviewer #1: I Don't Know

Reviewer #2: Yes

3. Have the authors made all data underlying the findings in their manuscript fully available?

Reviewer #1: Yes

Reviewer #2: Yes

4. Is the manuscript presented in an intelligible fashion and written in standard English?

Reviewer #1: Yes

Reviewer #2: Yes

5. Review Comments to the Author

Reviewer #1: In the manuscritp 'Curcumin improves D-galactose and Normal-aging Associated' the authors aimed to study the interaction of curcumin to GSTA1, GSTO1, KEAP1, BACE1, and MAOA proteins using Molecular docking and follow up if a curcumin treatment (maybe 77 or 80 days, this information is not clear) affects oxidative balance was also tested. The effect of curcumin on aging was associated with memory impairment in two animal models: D-Gal and normal aged mice.

I believe the manuscript needs corrections, my observations are to ask the authors the possibility of carrying out more experiments to try to answer the study's hypotheses. Furthermore, the work will need a variety of techniques and analysis to be considered publishable in PLOS One, in an attempt to bring more robust and less speculative evidence, as follows:

i) It is important to test the hypothesis that curcumin physically interacts with the proteins studied in the Molecular docking software. For this, it is essential to evaluate markers of these proteins in hippocampal samples, using WB and RT-PCR techniques.

ii) Knowing the distribution of these proteins in the hippocampal regions is highly significant, it would also be interesting to evaluate (in case of repetition of a new set of experiments) in CA1, CA2, CA3 and dentate gyrus, some antibodies used for WB can be used for immunohistochemistry;

iii) The experimental design (Figure 1) is missing some information and in the methodology. It is necessary to specify each step in the image, indicating the day of each of the arrows (example: day 0, day 7, day 30, day 77-78-79-80? final), look for experimental design models (use the software https://biorender.com/), as the one presented beyond of having poor resolution is visually depleted of elements and confusing;

iv) The statistical analysis must be carried out using the values of f as the standard;

v) Graphics don't usually have header title, correct that;

Reviewer #2: In presented article Rahman et. al investigates the influence of oral administration of curcumin on memory impairment in aging mice. The introduction to the article is well written and provides basic knowledge needed to understand the investigated features. It also exhaustively explains why the authors decided to investigate this subject. Materials and methods provide good basis for reconstitution of the experiments. However, the whole article requires some improvements prior to publishing.

My major concern is the language used in the results and discussion section. The authors repetitively say that cur or ast ‘elevates’, ‘increases’, ‘decreases’ even ‘restores’ the level of RT/FR/oxidative stress biomarkers in comparison to D-gal. While it seems rather as protection/prevention of the decrease. The authors describe that the level of some feature is A in control conditions, it’s decreased in d-gal treated mice but not in d-gal+cur/ast mice. It cannot be assumed that in the latter group of mice the level decreases and then is increased by cur/ast action as it was not tested. Also the expression “Control group exhibited X level of feature A which was decreased after administration of D-gal and increased after treatment with curcumin” suggests that the experiments were not done on separate groups of animals but rather the same mice were investigated in control state, then injected with D-gal and tested and, finally, fed with curcumin and tested. Such language is used throughout the whole results section.

In the section 3.1 the authors present results of passive avoidance test after 24h and 48h. The results are similar at both time-points and presentation of 48h doesn’t introduce any new information. Consider moving it to the supplement.

In the section 4.3 authors write that “curcumin’s activity minimized the aging-induced oxidative burden”. I wouldn’t agree that the oxidative burden is induced by aging, but it is rather associated with aging.

The title of section 4.4 is a bit misleading. From what I understood the authors did in silico experiments of molecular docking to predict interactions while the title firmly states that such interaction occurs.

The last paragraph of section 4.4 is just a recapitulation of what was already said in the introduction.

6. PLOS authors have the option to publish the peer review history of their article (what does this mean?). If published, this will include your full peer review and any attached files.

Reviewer #1: No

Reviewer #2: **Yes: **Wioleta Grabowska-Pyrzewicz PhD

---

## [Author Response · Author response to Decision Letter 0]

25 May 2022

Response to Reviewers

We thank the reviewers for reading the manuscript carefully and providing productive comments. We have subsequently revised our manuscript titled "Curcumin improves D-galactose and Normal-aging Associated Memory Impairment in Mice: in vivo and in silico-Based Studies”

The comments made by each reviewer are first copied below (underlined) and based on these our responses were provided at the end of the reviewer’s individual remarks. Page and line numbers in our reply refer to those in the 'Revised Manuscript with Track Changes.' 

Response: In response to this comment, we have revised our manuscripts to meet PLOS ONE's style requirements.

2. Upon re-submitting your revised manuscript, please upload your study’s minimal underlying data set as either Supporting Information files or to a stable, public repository and include the relevant URLs, DOIs, or accession numbers within your revised cover letter. 

Response: All relevant data are within the manuscript

3. "The funders had no role in study design, data collection and analysis, decision to publish, or preparation of the manuscript." At this time, please address the following queries

Response: In response to this comment, we have mentioned that this work was partially supported by the research grant provided by the CTRG-North South University 2019–2020 and the National Institute of Science and Technology (NST- 2020). 

1. Is the manuscript technically sound, and do the data support the conclusions?

Reviewer #1: No

Reviewer #2: Yes

2. Has the statistical analysis been performed appropriately and rigorously?

Reviewer #1: I Don't Know

Reviewer #2: Yes

3. Have the authors made all data underlying the findings in their manuscript fully available?

Reviewer #1: Yes

Reviewer #2: Yes

4. Is the manuscript presented in an intelligible fashion and written in standard English?

Reviewer #1: Yes

Reviewer #2: Yes

5. Review Comments to the Author

Reviewer #1: In the manuscript 'Curcumin improves D-galactose and Normal-aging Associated' the authors aimed to study the interaction of curcumin to GSTA1, GSTO1, KEAP1, BACE1, and MAOA proteins using Molecular docking and follow up if a curcumin treatment (maybe 77 or 80 days, this information is not clear) affects oxidative balance was also tested. The effect of curcumin on aging was associated with memory impairment in two animal models: D-Gal and normal aged mice.

I believe the manuscript needs corrections, my observations are to ask the authors the possibility of carrying out more experiments to try to answer the study's hypotheses. Furthermore, the work will need a variety of techniques and analysis to be considered publishable in PLOS One, in an attempt to bring more robust and less speculative evidence, as follows:

Response: We thank Reviewer#1 for precisely summarizing our findings. We have mentioned the curcumin treatment duration in Fig 1.

i) It is important to test the hypothesis that curcumin physically interacts with the proteins studied in the Molecular docking software. For this, it is essential to evaluate markers of these proteins in hippocampal samples, using WB and RT-PCR techniques.

Response: We again thank Reviewer#1 for the valuable remarks. The in-silico studies clearly showed that GSTA1, GSTO1, KEAP1, BACE1, and MAOA proteins interacted with the curcumin and supported our hypothesis. We agree that the WB and RT-PCR techniques would help us potentiate our findings. We did not adopt those techniques in our study because of not the availability of these facilities in our lab at present. 

ii) Knowing the distribution of these proteins in the hippocampal regions is highly significant, it would also be interesting to evaluate (in case of repetition of a new set of experiments) in CA1, CA2, CA3 and dentate gyrus, some antibodies used for WB can be used for immunohistochemistry;

Response: We appreciated Reviewer#1 for the valuable remarks. We have added some lines (Page 21 and 22; lines 592-595; 602-604; 615-616; 618-619 in the file with changes red color marked).

GSTA1, GSTO1, and KEAP1 are abundantly present in the hippocampus, a brain region crucial for hippocampus-dependent learning tasks [74, 75]. Western blot analysis showed that the upregulation of GSTA1 in the CA1 area [76] and the downregulation of GSTO1 in the hippocampus were linked to cognitive impairment [77] commonly seen in aging animals. The Keap1-Nrf2 system plays a crucial role in regulating oxidative stress-mediated disorders [80]. A western blot analysis found a lower expression of KEAP1 in CA3 and dentate gyrus of the hippocampus under oxidative conditions [80]. Immunohistochemical studies showed a high level of MAOA in the CA3 area of the hippocampus, an important region sensitive for brain aging [84]. BACE1 is widely distributed in CA1 and CA3 areas, and the absence of this protein is responsible for the altered level of synaptic plasticity in aging mice [86].

References

74. Rahman MA, Tanaka N, Usui K, Kawahara S. Role of Muscarinic Acetylcholine Receptors in Serial Feature-Positive Discrimination Task during Eyeblink Conditioning in Mice. Plos one. 2016 Jan 25;11(1):e0147572.

75. Rahman MA, Tanaka N, Nuruzzaman M, DebNath S, Kawahara S. Blockade of the M1 muscarinic acetylcholine receptors impairs eyeblink serial feature-positive discrimination learning in mice. PloS one. 2020 Aug 13;15(8):e0237451.

76. Landfield P, Blalock E, Chan KC, Fossex T, inventors; University of Kentucky Research Foundation, assignee. Gene expression profile biomarkers and therapeutic targets for brain aging and age-related cognitive impairment. United States patent application US 10/486,706. 2005 Mar 31.

77. Lee JM, Lee JH, Song MK, Kim YJ. NXP032 Ameliorates Aging-Induced Oxidative Stress and Cognitive Impairment in Mice through Activation of Nrf2 Signaling. Antioxidants. 2022 Jan;11(1):130.

80. Yao W, Zhang JC, Ishima T, Dong C, Yang C, Ren Q, Ma M, Han M, Wu J, Suganuma H, Ushida Y. Role of Keap1-Nrf2 signaling in depression and dietary intake of glucoraphanin confers stress resilience in mice. Scientific reports. 2016 Jul 29;6(1):1-3.

84. Price D, Gaspar P, Cases O. Developmental expression of monoamine oxidases A and B in the central and peripheral nervous systems of the mouse. Journal of Comparative Neurology. 2002 Jan 21;442(4):331-47.

86. Laird FM, Cai H, Savonenko AV, Farah MH, He K, Melnikova T, Wen H, Chiang HC, Xu G, Koliatsos VE, Borchelt DR. BACE1, a major determinant of selective vulnerability of the brain to amyloid-β amyloidogenesis, is essential for cognitive, emotional, and synaptic functions. Journal of Neuroscience. 2005 Dec 14;25(50):11693-709.

iii) The experimental design (Figure 1) is missing some information and in the methodology. It is necessary to specify each step in the image, indicating the day of each of the arrows (example: day 0, day 7, day 30, day 77-78-79-80? final), look for experimental design models (use the software https://biorender.com/), as the one presented beyond of having poor resolution is visually depleted of elements and confusing;

Response: We again thank Reviewer#1 for the valuable suggestions. We have changed Fig.1 accordingly.

iv) The statistical analysis must be carried out using the values of f as the standard;

Response: We acknowledge Reviewer#1 for his remarks. We have incorporated the F values in our entire result section.

v) Graphics don't usually have header title, correct that;

Response: We again admit the comments of Reviewer#1. We have deleted the header title from the figures.

Reviewer #2: In presented article Rahman et. al investigates the influence of oral administration of curcumin on memory impairment in aging mice. The introduction to the article is well written and provides basic knowledge needed to understand the investigated features. It also exhaustively explains why the authors decided to investigate this subject. Materials and methods provide good basis for reconstitution of the experiments. However, the whole article requires some improvements prior to publishing.

Response: We thank Reviewer#2 for clearly summarizing our findings. 

My major concern is the language used in the results and discussion section. The authors repetitively say that cur or ast ‘elevates’, ‘increases’, ‘decreases’ even ‘restores’ the level of RT/FR/oxidative stress biomarkers in comparison to D-gal. While it seems rather as protection/prevention of the decrease. 

Response: We appreciate the valuable suggestions of Reviewer 2. We have used protection/prevention of the decrease in place of elevates’, ‘increases’, ‘decrease’. (Page 12: line 312; Page 13: line 343, 350, 354, 355, 365; Page 14: line 381, 384; Page 15: 401, 407, 423; Page 16: 428, 438, 449; Page 17: 462, 467; Page 19: 525, 546, 564, 576, 580, in the file with changes red color marked).

The authors describe that the level of some feature is A in control conditions, it’s decreased in d-gal treated mice but not in d-gal+cur/ast mice. It cannot be assumed that in the latter group of mice the level decreases and then is increased by cur/ast action as it was not tested. Also the expression “Control group exhibited X level of feature A which was decreased after administration of D-gal and increased after treatment with curcumin” suggests that the experiments were not done on separate groups of animals but rather the same mice were investigated in control state, then injected with D-gal and tested and, finally, fed with curcumin and tested. Such language is used throughout the whole results section.

Response: We again acknowledge Reviewer 2 for the valuable findings. We have changed the language accordingly in the result section (3.1 to 3.3).

In the section 3.1 the authors present results of passive avoidance test after 24h and 48h. The results are similar at both time-points and presentation of 48h doesn’t introduce any new information. Consider moving it to the supplement.

Response: We are thankful to Reviewer 2 for the valuable remarks. We have shifted the fig of 48 hours in the supplementary section (see SN 1). Also, we have rewritten a separate figure legend of SN1 (page 33, line 970-973 in red color marked).

In the section 4.3 authors write that “curcumin’s activity minimized the aging-induced oxidative burden”. I wouldn’t agree that the oxidative burden is induced by aging, but it is rather associated with aging.

Response: We again appreciate Reviewer 2 for the suggestion. We have changed the language accordingly in the result section (page 21: line 578 in the file with changes red color marked).

The title of section 4.4 is a bit misleading. From what I understood the authors did in silico experiments of molecular docking to predict interactions while the title firmly states that such interaction occurs.

Response: We are thankful to Reviewer 2 for the valuable remarks. We have changed the language accordingly in the title (page 21: line 585 in the file with changes red color marked). 

The last paragraph of section 4.4 is just a recapitulation of what was already said in the introduction.

Response: We appreciate Reviewer 2 for the valuable findings. We have changed the language accordingly (page 22: line 612-613, 632, in the file with changes red color marked).

---

## [Editor Report · Decision Letter 1]

6 Jun 2022

Curcumin improves D-galactose and Normal-aging Associated Memory Impairment in Mice: in vivo and in silico-Based Studies

PONE-D-21-30364R1

Dear Dr. Rahman,

We’re pleased to inform you that your manuscript has been judged scientifically suitable for publication and will be formally accepted for publication once it meets all outstanding technical requirements.

Kind regards,

Michelle Melgarejo da Rosa

Academic Editor

PLOS ONE
---

## [Editor Report · Acceptance letter]

20 Jun 2022

PONE-D-21-30364R1 

Curcumin improves D-galactose and Normal-aging Associated Memory Impairment in Mice: in vivo and in silico-Based Studies 

Dear Dr. Rahman:

I'm pleased to inform you that your manuscript has been deemed suitable for publication in PLOS ONE. Congratulations! Your manuscript is now with our production department. 

Kind regards, 

on behalf of

Dr. Michelle Melgarejo da Rosa 

Academic Editor

PLOS ONE